# ADF and cofilin-1 collaborate to promote cortical actin flow and the leader bleb-based migration of confined cells

Maria F Ullo, Jeremy S Logue*

Department of Regenerative and Cancer Cell Biology, Albany Medical College, Albany, United States

**Abstract** Melanoma cells have been shown to undergo fast amoeboid (leader bleb-based) migration, requiring a single large bleb for migration. In leader blebs, is a rapid flow of cortical actin that drives the cell forward. Using RNAi, we find that co-depleting cofilin-1 and actin depolymerizing factor (ADF) led to a large increase in cortical actin, suggesting that both proteins regulate cortical actin. Furthermore, severing factors can promote contractility through the regulation of actin architecture. However, RNAi of cofilin-1 but not ADF led to a significant decrease in cell stiffness. We found cofilin-1 to be enriched at leader bleb necks, whereas RNAi of cofilin-1 and ADF reduced bleb sizes and the frequency of motile cells. Strikingly, cells without cofilin-1 and ADF had blebs with abnormally long necks. Many of these blebs failed to retract and displayed slow actin turnover. Collectively, our data identifies cofilin-1 and ADF as actin remodeling factors required for fast amoeboid migration.

*For correspondence:
loguej@mail.amc.edu

Competing interests: The authors declare that no competing interests exist.

## Introduction

Cell migration requires tight spatiotemporal control of the filamentous-actin (F-actin) cytoskeleton. For mesenchymal migration, actin assembly/disassembly and myosin contraction must occur within specific regions of the cell (*Pollard and Borisy, 2003*). Relative to other, recently described modes of migration, the mechanisms by which mesenchymal cells coordinate these processes are reasonably well understood. Whereas, in amoeboid cells, which migrate using intracellular-driven protrusions of the plasma membrane (PM) or blebs, the mechanisms conferring spatiotemporal control of the F-actin cytoskeleton are not well known.

Within tissues, cells encounter a variety of physicochemical environments. We and others discovered that in response to tissue confinement, cancer cells frequently undergo phenotypic transitions, including to what has been termed 'fast amoeboid migration' (*Logue et al., 2015*; *Liu et al., 2015*; *Ruprecht et al., 2015*; *Bergert et al., 2015*; *Mistriotis et al., 2019*; *Wisniewski et al., 2020*). A hallmark of amoeboid migration is the presence of blebs, which form as a result of PM-cortical actin separation (*Charras et al., 2008*). Typically, blebs are rapidly retracted following the reassembly of cortical actin on bleb membranes and the recruitment of myosin (*Charras et al., 2008*). However, during fast amoeboid migration, cells form a very large and stable bleb. Within these blebs, is a cortical actin network that flows from the bleb tip to the neck, which separates leader blebs from the cell body (*Logue et al., 2015*; *Liu et al., 2015*; *Ruprecht et al., 2015*; *Bergert et al., 2015*). Together with non-specific friction, flowing cortical actin provides the motive force for cell movement (*Ruprecht et al., 2015*). Accordingly, we simultaneously termed this mode of migration, leader bleb-based migration (LBBM) (*Logue et al., 2015*). Because migration plasticity is thought to be a major contributor of metastasis, our aim here is to identify the essential factors required for the rapid cortical actin flow in leader blebs.

Previous studies have demonstrated that myosin is enriched at the neck of leader blebs. Indeed, the myosin at this location has been shown to be required for rapid cortical actin flow (*Liu et al., 2015*; *Ruprecht et al., 2015*; *Bergert et al., 2015*). However, what drives the disassembly of F-actin at the neck to replenish the pool of G-actin at the leader bleb tip is unknown. Thus, we hypothesize that actin disassembly factors within leader blebs play a pivotal role in this process. Similarly, in mesenchymal cells, retrograde actin flow is driven by both myosin contraction and actin disassembly, the latter of which is accelerated by the actin depolymerizing factor (ADF)/cofilin family of severing factors (*Bravo-Cordero et al., 2013*). Severing depends on ADF/cofilin inducing a structural change within the actin filament, an activity known to be repressed by LIM kinase (LIMK) phosphorylation on Ser 3 (*Yang et al., 1998*). However, the consequence of actin severing is variable, depending on the availability of barbed end capping proteins. At a sufficiently high level, new barbed ends are capped, promoting actin disassembly from new pointed ends (*Wioland et al., 2017*). Conversely, if left uncapped, increased polymerization from new barbed ends may occur (*Wioland et al., 2017*). The effects of ADF/cofilin on actin severing is also concentration dependent, where low and high levels of ADF/cofilin sever and stabilize actin filaments, respectively (*Andrianantoandro and Pollard, 2006*). Additionally, by competing for F-actin binding (e.g., Arp2/3 and other mechanisms, ADF/cofilin can regulate actin architecture (*Chan et al., 2009*; *Gressin et al., 2015*). The effects of ADF/cofilin on cortical actin are even less certain, as the focus of many studies has been on the actin within lamellipodia. Although Wiggan et al. have previously reported bleb defects in HeLa cells depleted of actin severing factors, the role of ADF and cofilin-1 in regulating the cortical actomyosin flow in fast amoeboid cells has not been determined (*Wiggan et al., 2012*).

By combining an in vitro assay for the precise confinement of cells with quantitative imaging approaches, we report that ADF and cofilin-1, together, are required for the rapid disassembly of incoming cortical actin at leader bleb necks. Under conditions of confinement, we find that melanoma and lung adenocarcinoma cells depleted of both proteins display dramatic defects in bleb morphology and dynamics. Consequently, cells without ADF and cofilin-1 cannot undergo LBBM. Thus, we reveal unanticipated roles for ADF and cofilin-1 in driving confined (leader bleb-based) migration.

## Results

### ADF and cofilin-1 are required for LBBM

Using our previously described approach for cell confinement, which involves placing cells under a slab of PDMS held at a defined height (~3 μm) above cover glass, cancer cells will switch to LBBM (*Figure 1A* and *Video 1*; *Logue et al., 2018*). Moreover, within leader blebs, we find a rapid flow of cortical actin, which together with non-specific friction, provides the motive force for cell movement (*Figure 1B* and *Video 2*; *Bergert et al., 2015*). As indicated by an enrichment in EGFP tagged regulatory light chain (EGFP-RLC), we observe a concentration of myosin at the leader bleb neck that separates the leader bleb from the cell body (*Figure 1C* and *Video 2*; *Liu et al., 2015*; *Ruprecht et al., 2015*; *Bergert et al., 2015*). In concert with myosin, we wondered if the cortical actin flow in leader blebs requires the action of specific actin disassembly factors.

In addition to being frequently up-regulated in cancer, the ADF/cofilin family of actin severing factors are known to be essential for actin turnover in mesenchymal cells (*Bravo-Cordero et al., 2013*; *Bracalente et al., 2018*). Therefore, we set out to determine if ADF and/or cofilin-1 are important for LBBM. In melanoma A375 cells, a widely used cell line for the study of amoeboid migration, both ADF and cofilin-1 are expressed, with cofilin-1 mRNA levels being threefold higher (*Figure 1—figure supplement 1B*). Using A375 cells, we depleted cells of ADF and cofilin-1 alone and together by RNAi (*Figure 1D* and *Figure 1—figure supplement 1A*). By manually tracking the movement of cells over time, we found that cells depleted of cofilin-1 were significantly less motile (*Figure 1E–H* and *Figure 1—figure supplement 1C*). Additionally, the adhesive transmigration of cells through small pores was hindered after depleting cofilin-1 (*Figure 1—figure supplement 1D*). Depleting cells of ADF led to a slight reduction in the number of highly motile cells (*Figure 1E–H* and *Figure 1—figure supplement 1C*). Strikingly, depleting cells of both ADF and cofilin-1 appeared to have an additive effect on reducing LBBM, whereas the adhesive transmigration of cells through small pores did not display this additive behavior and was not affected by ADF depletion

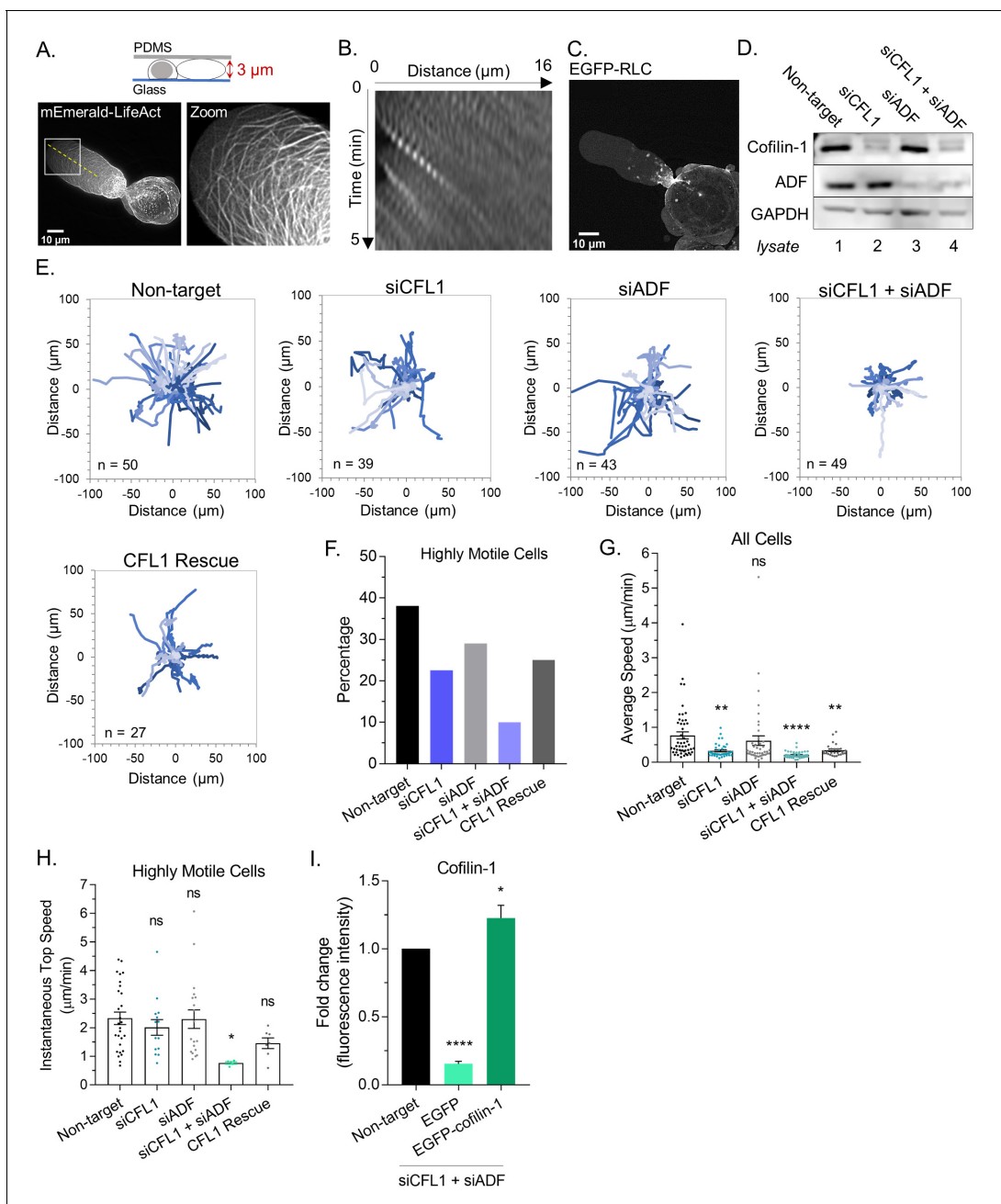

**Figure 1.** ADF and cofilin-1 are required for leader bleb-based migration. (**A**) Ventral Z-section of a melanoma A375-M2 cell, which has been confined down to 3 μm, with mEmerald-LifeAct. (**B**) Kymograph from (A; dashed line), showing cortical actin flow. (**C**) Ventral Z-section of a melanoma A375-M2 cells, which has been confined down to 3 μm, with EGFP tagged regulatory light chain (EGFP-RLC). (**D**) Western blot confirming CFL1, actin depolymerizing factor (ADF), and ADF + CFL1 RNAi in melanoma A375-M2 cells. (**E**) Individual cell migration tracks (plot of origin) for non-targeting, CFL1, ADF, and CFL1 + ADF RNAi cells, as well as CFL1 + ADF RNAi cells rescued by transfection with EGFP-cofilin-1 plasmid. In each, cells were tracked over a period of 5 hr. Relative y (μm) and relative × (μm) are shown in each. (**F**) Percentage of highly motile cells from (**E**). Cells that traveled a distance equivalent to at least one cell length over the course of the 5 hr time-lapse were classified as highly motile. (**G**) Average speed (μm/min) from cells in (E; mean ± SEM). Statistical significance was determined by one-way ANOVA and a Dunnett's post hoc test. (**H**) Instantaneous top speed (μm/min) for highly motile cells in (E; mean ± SEM). (**I**) Cofilin-1 levels (fold change; fluorescence intensity) of adhered RNAi cells by immunofluorescence confirming rescue by transfection with EGFP-cofilin-1 or not rescued with EGFP. Statistical significance was determined by an unpaired one-sample t-test. All data are representative of at least three independent experiments. *p ≤ 0.05, **p ≤ 0.01, ***p ≤ 0.001, and ****p ≤ 0.0001.

The online version of this article includes the following source data and figure supplement(s) for figure 1:

**Source data 1.** Raw data from manual tracking.

**Figure supplement 1.** Protein levels and migration parameters.

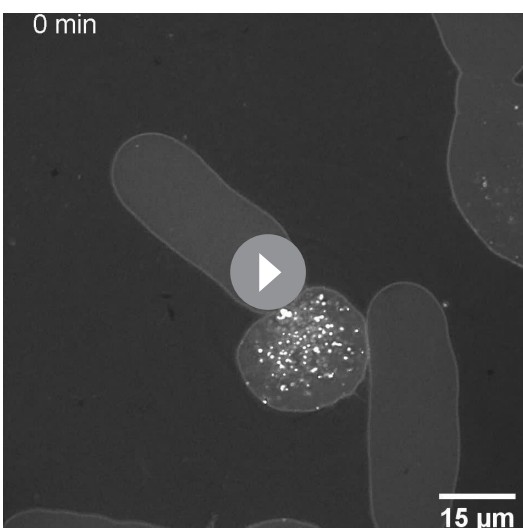

**Video 1.** Time-lapse imaging of melanoma A375-M2 cells confined down to 3 μm with far-red plasma membrane dye.
https://elifesciences.org/articles/67856#video1

(*Figure 1E–H* and *Figure 1—figure supplement 1D*). These results suggest that these proteins play non-overlapping roles during LBBM (*Figure 1E–H*). In agreement with this concept, transfection of EGFP-cofilin-1 into cells depleted of both proteins was insufficient to restore LBBM (*Figure 1E–I*).

## Together, ADF and cofilin-1 are required to retract blebs

Subsequently, we wanted to know what is responsible for the decrease in LBBM upon depleting ADF and/or cofilin-1. Initially, we analyzed the area of the largest bleb (i.e., leader bleb), relative to the cell body, in cells depleted of ADF and/or cofilin-1. This analysis revealed that depleting cells of cofilin-1 reduced the area of the largest bleb by ~15% of control while depleting cells of both ADF and cofilin-1 reduced the area of the largest bleb by ~40% of control (non-targeting; *Figure 2A–B* and *Videos 3–5*). The area of all blebs was similarly reduced (*Figure 2C*). Strikingly, in cells depleted of both ADF and cofilin-1, ~30% of cells displayed blebs with elongated necks (*Figure 2A,D* and *Video 6*). Detailed analyses of these cells revealed that ~60% display blebs that never retract into the cell body (*Figure 2E*). Moreover, we found this effect to not be specific to melanoma A375 cells, as depleting lung adenocarcinoma A549 cells of both ADF and cofilin-1 similarly resulted in the elongation of bleb necks (*Figure 2F–G*). In melanoma A375 cells, depleting both ADF and cofilin-1 had an additive effect on reducing the rate of bleb retraction, which points to these proteins having non-overlapping functions specifically at bleb necks (*Figure 2H*). In support of this role, we determined the location of EGFP-cofilin-1 in confined cells. Our initial efforts were unsuccessful because of a high degree of soluble (i.e., unbound to F-actin) protein, but in cells depleted of endogenous ADF and cofilin-1 we could detect an enrichment of EGFP-cofilin-1 at leader bleb necks (*Figure 2I*; *arrow*).

## ADF and cofilin-1 rapidly disassemble cortical actin

Thus far, our results suggest that ADF and cofilin-1 are important for the rapid turnover of cortical actin at bleb necks. In agreement with this concept, a large concentration of F-actin is found at the necks of blebs in cells depleted of ADF and cofilin-1 (*Figure 3A*). In order to more directly test the role of ADF and cofilin-1 in regulating cortical actin, we turned to freshly trypsinized (spherical) cells. Previous work by us and others has shown that the properties of the cortical actin network in spherical cells correlate with LBBM (*Logue et al., 2015*; *Liu et al., 2015*; *Bergert et al., 2012*). In spherical cells, actin is predominantly cortical and endogenous cofilin-1 is diffuse throughout the cytoplasm with some enrichment at the cell periphery (*Figure 3B*). By

**Video 2.** Time-lapse imaging of a melanoma A375-M2 cell confined down to 3 μm with mScarlet-LifeAct and EGFP tagged regulatory light chain (EGFP-RLC).
https://elifesciences.org/articles/67856#video2

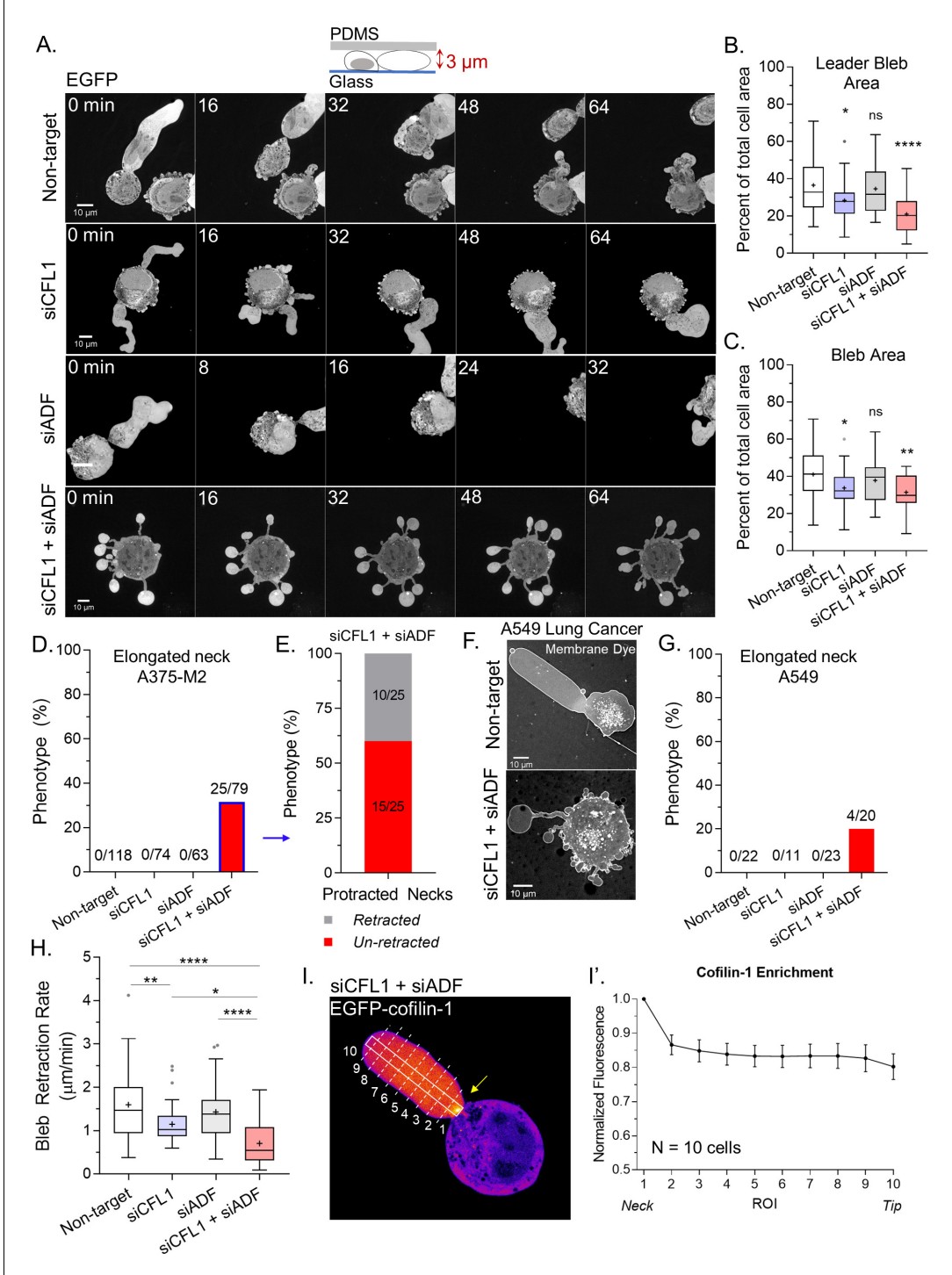

**Figure 2.** Together, actin depolymerizing factor (ADF) and cofilin-1 are required to retract blebs. (**A**) Montage of non-targeting, CFL1, ADF, and CFL1 + ADF RNAi with EGFP alone (volume marker) in melanoma A375-M2 cells. (**B–C**) Quantitation of area for leader (**A**) and all blebs (**B**) after non-targeting, CFL1, ADF, and ADF + CFL1 RNAi. Statistical significance was determined by one-way ANOVA and a Dunnet's post hoc test. (**D**) Percent of non-targeting, CFL1, ADF, and CFL1 + ADF RNAi cells with elongated bleb necks. (**E**) Percent of ADF + CFL1 RNAi cells from (**D**) with elongated bleb necks that retract vs. un-retracted. (**F–G**) Lung adenocarcinoma A549 cells after non-targeting and CFL1 + ADF RNAi stained with a far-red fluorescent membrane dye (**F**). Percent of non-targeting, CFL1, ADF, and CFL1 + ADF RNAi cells with elongated bleb necks (**G**). (**H**) Bleb retraction rates for non-targeting (45 blebs; 26 cells), CFL1 (40 blebs; 20 cells), ADF (48 blebs; 30 cells), and CFL1 + ADF RNAi (38 blebs; 23 cells). Statistical significance was determined by one-way ANOVA and a Dunnet's post hoc test. (**I**) EGFP-cofilin-1 localization in an A375-M2 cell confined down to 3 μm. Arrow points to an enrichment of cofilin-1 at the leader bleb neck. (**I'**) Regional analysis of EGFP-cofilin-1 average fluorescence intensity in ROIs sampled from bleb neck

*Figure 2 continued on next page*

*Figure 2 continued*

to tip (mean ± SEM). Representative regions taken within white box and dashed lines in (I). All data are representative of at least three independent experiments. *p ≤ 0.05, **p ≤ 0.01, ***p ≤ 0.001, and ****p ≤ 0.0001.

The online version of this article includes the following source data for figure 2:

**Source data 1.** Raw leader bleb and bleb area measurements.

combining the specificity of phalloidin for F-actin with flow cytometry, we then determined how the level of cortical actin is affected by ADF and/or cofilin-1 depletion. In cells depleted of cofilin-1, F-actin levels were increased by ~10%, whereas depletion of ADF did not lead to a significant change in the level of F-actin (*Figure 3C*). Interestingly, depleting ADF with cofilin-1 had the largest effect, increasing F-actin levels by ~30% (*Figure 3C*). Serine 3 of cofilin-1 is phosphorylated by LIMK, inhibiting its severing activity (*Yang et al., 1998*). In cells depleted of endogenous ADF and cofilin-1, transfection of EGFP-cofilin-1 reduced the level of F-actin to levels similar to control (non-target; *Figure 3D*). Similarly, F-actin was restored to near control levels by transfection of EGFP-cofilin-1 (S3A; constitutively active), whereas transfection of EGFP-cofilin-1 (S3E; constitutively inactive) led to an increased level of F-actin (*Figure 3D*). Additionally, in comparison to EGFP alone, increasing levels of EGFP-cofilin-1 correlated with reductions in F-actin (*Figure 3E*). Thus, cortical actin levels are regulated by ADF and cofilin-1 severing.

Many studies have shown that cofilin-1 severing can result in distinct outcomes. In lamellipodia, cofilin-1 severing promotes actin polymerization at new barbed ends, whereas in the lamella, actin is de-polymerized at new pointed ends (*Bravo-Cordero et al., 2013*). The prevalence of each outcome has been proposed to depend on the concentration of capping proteins (*Wioland et al., 2017*). Therefore, we determined the relative level of barbed ends within the cortical actin network of freshly trypsinized (spherical) cells. The treatment of cells with cytochalasin B (which blocks G-actin from binding the barbed end) confirmed the specificity of the approach (*Figure 3—figure supplement 1A*). In cells depleted of cofilin-1, we observed a more than ~50% increase the level of cortical barbed ends, whereas depleting ADF marginally increased the level of cortical barbed ends (*Figure 3G*). Depleting cells of both proteins appeared to have an additive effect, increasing the level of cortical barbed ends by ~100% (*Figure 3G*). However, in cells depleted of ADF and cofilin-1, transfection of EGFP-cofilin-1 was sufficient to restore cortical barbed ends to a level similar to control (*Figure 3H*). Collectively, these results suggest that ADF and cofilin-1 regulate cortical actin levels by promoting the de-polymerization of actin at newly formed pointed ends.

## Rapid cortical actin flow requires ADF and cofilin-1 severing at leader bleb necks

Lasting changes in the F/G-actin ratio, such as occurs during RNAi of ADF and/or cofilin-1, can have a range of effects on cellular physiology. Therefore, using a previously described Super-Nova-cofilin-1 construct, we performed chromophore assisted light inactivation (CALI) for establishing the direct effects of cofilin-1 depletion (*Vitriol et al., 2013*; *Takemoto et al., 2013*). In cells depleted of ADF and cofilin-1, Super-Nova-cofilin-1 is predominantly observed at leader bleb necks (*Figure 4A*). After 1 min of intense red light irradiation, we found that ~50% of SuperNova-cofilin-1 is destroyed, as indicated by a reduction in red fluorescence (*Figure 4A*). Using LifeAct-mEmerald for monitoring actin dynamics, we observed an accumulation of actin and the elongation of leader bleb necks within

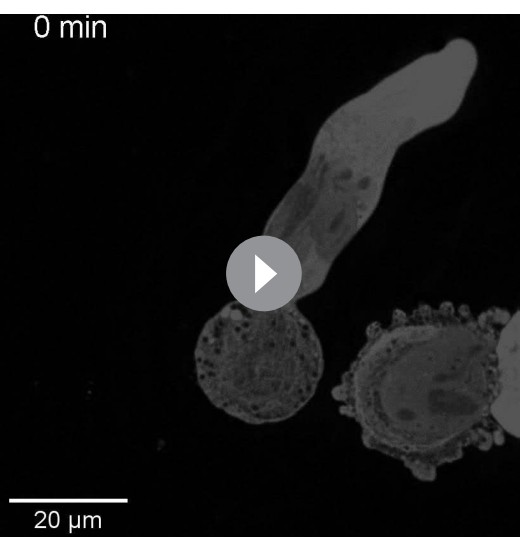

**Video 3.** Time-lapse imaging of a melanoma A375-M2 cell confined down to 3 μm with the volume marker, mScarlet, after control (non-targeting) RNAi. https://elifesciences.org/articles/67856#video3

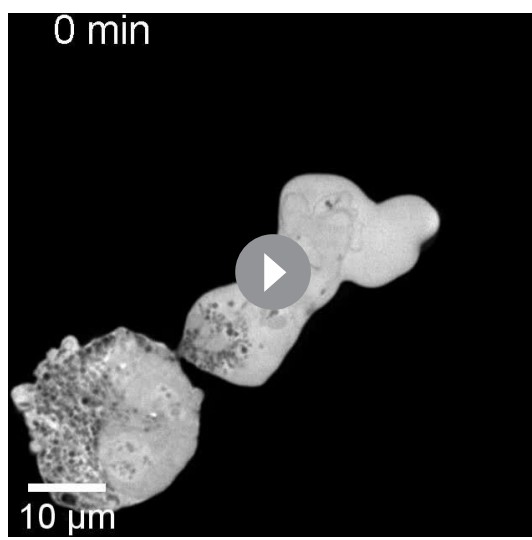

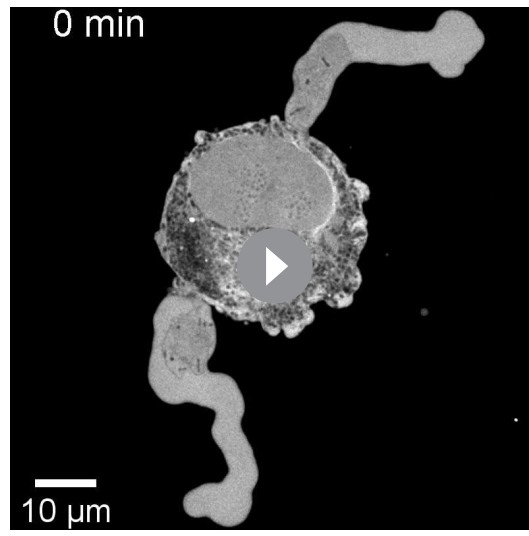

**Video 4.** Time-lapse imaging of a melanoma A375-M2 cell confined down to 3 μm with the volume marker, EGFP, after RNAi of actin depolymerizing factor (ADF) alone.
https://elifesciences.org/articles/67856#video4

**Video 5.** Time-lapse imaging of a melanoma A375-M2 cell confined down to 3 μm with the volume marker, mScarlet, after RNAi of CFL1 alone.
https://elifesciences.org/articles/67856#video5

minutes of inactivating cofilin-1 (*Figure 4B* and *Video 7*). Moreover, after inactivating Super-Nova-cofilin-1, cortical actin flow rates in leader blebs were reduced by ~50%, whereas irradiation of SuperNova alone did not change actin flow rates (*Figure 4C–D* and *Figure 4—figure supplement 1A*). Thus, as demonstrated by CALI, cofilin-1 directly regulates the cortical actin in leader blebs.

We then determined rates of actin turnover at leader bleb necks. For this, we used a version of LifeAct tagged with the photoactivable (green/red) fluorescent protein, mEos, for performing fluorescence loss after photoactivation assays. Using a 405 nm laser, a pool of mEos-LifeAct was photo-activated at leader bleb necks. Subsequently, fluorescence decay at leader bleb necks was measured and used for determining rates of actin turnover (*Figure 4E*). However, as our measurements will be a composite of the actin turnover rate and the binding kinetics of LifeAct, these data should be strictly viewed as apparent rates of actin turnover. To demonstrate proof of concept, curves were fit with a single phase decay, yielding an R-squared value of 0.8348 (*Figure 4F*). In contrast, a rapid decline in red fluorescence was not observed in paraformaldehyde treated cells (*Figure 4—figure supplement 1B*). In control (non-target) cells, we found the actin at leader bleb necks to be rapidly turned over ($t_{1/2}$; *Figure 4G*). While the rate of actin turnover in cells depleted of cofilin-1 or ADF alone was similar to control (non-target), depleting both ADF and cofilin-1 led to a significant decrease in the actin turnover rate ($t_{1/2}$; *Figure 4G*). These data suggest that ADF and cofilin-1, together, are critical for the rapid turnover of cortical actin at leader bleb necks.

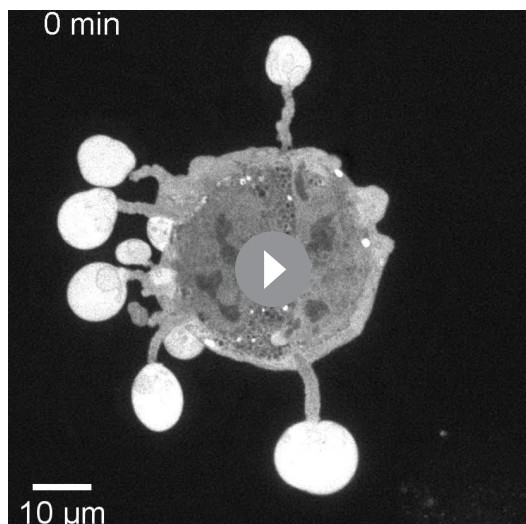

**Video 6.** Time-lapse imaging of a melanoma A375-M2 cell confined down to 3 μm with the volume marker, EGFP, after RNAi of CFL1 + actin depolymerizing factor (ADF).
https://elifesciences.org/articles/67856#video6

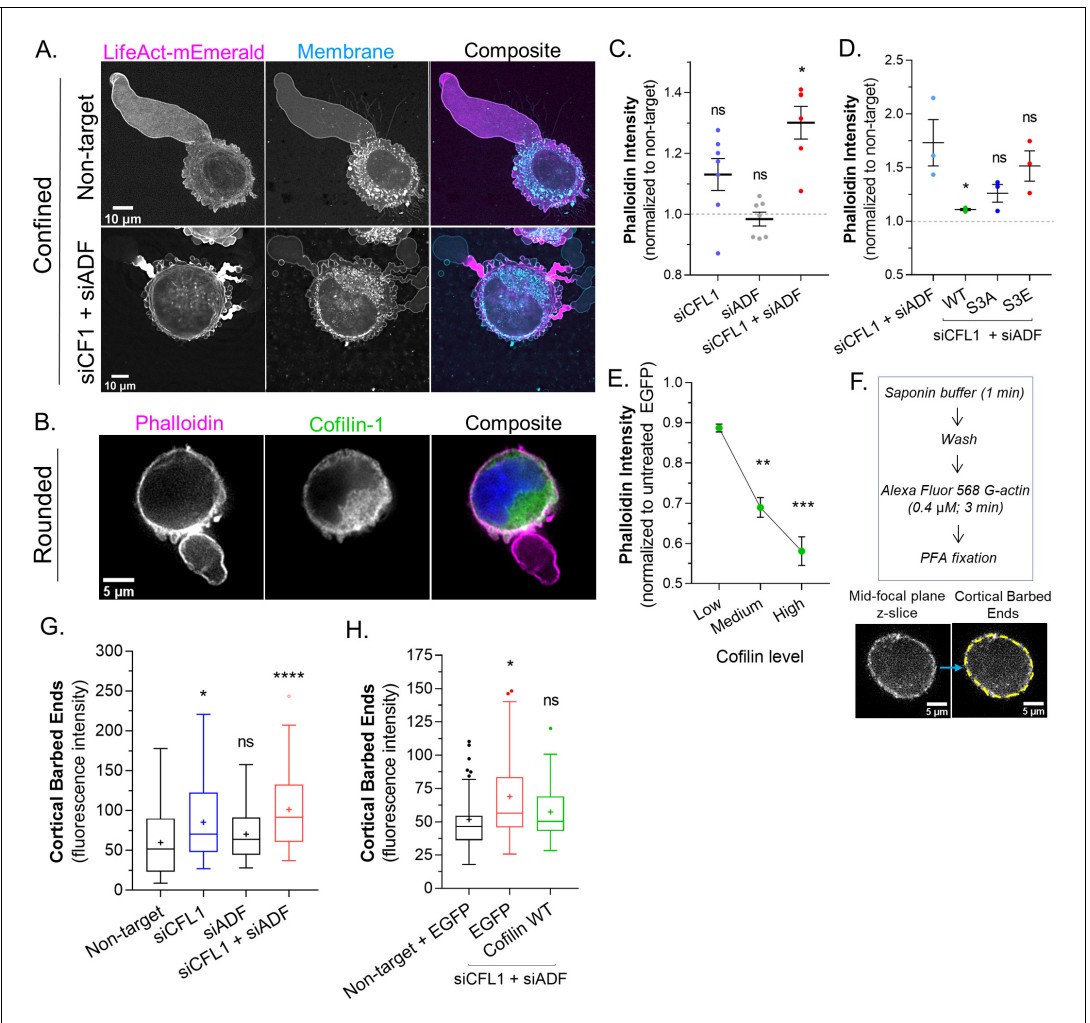

**Figure 3.** Actin depolymerizing factor ( ADF) and cofilin-1 rapidly disassemble cortical actin. (**A**) mEmerald-LifeAct and far-red fluorescent membrane dye in cells after non-targeting and CFL1 + ADF RNAi. (**B**) Cells freshly plated on poly-L-lysine coated cover glass stained for endogenous cofilin-1 and filamentous-actin (F-actin) (phalloidin). (**C**) F-actin levels (normalized to non-target; mean ± SEM) after CFL1, ADF, and CFL1 + ADF RNAi in trypsinized (spherical) cells, as determined by flow cytometry. Statistical significance was determined by one-way ANOVA and a Dunnet's post hoc test. (**D**) F-actin levels (normalized to non-target; mean ± SEM) after CFL1 + ADF RNAi, as well as after CFL1 + ADF RNAi with EGFP-cofilin-1 WT, S3A, or S3E, as determined by flow cytometry. Statistical significance was determined by one-way ANOVA and a Dunnet's post hoc test. (**E**) F-actin level (normalized to EGFP alone; mean ± SEM) as a function of increasing EGFP-cofilin-1 in cells depleted of endogenous cofilin-1 and ADF by RNAi, as determined by flow cytometry. Statistical significance was determined by one-way ANOVA and a Dunnet's post hoc test. (**F**) *Top*, barbed end assay workflow. *Bottom*, representative image of a freshly plated (spherical) cell subjected to the barbed end assay. (**G**) As shown in (F; *bottom*), the level of cortical barbed ends was measured in cells after non-targeting (71 cells), CFL1 (53 cells), ADF (47 cells), and CFL1 + ADF RNAi (83 cells). Statistical significance was determined by one-way ANOVA and a Dunnet's post hoc test. (**H**) As shown in (F; *bottom*), the level of cortical barbed ends was measured in cells with non-targeting and EGFP (42 cells), as well as after CFL1 + ADF RNAi with EGFP (32 cells) or EGFP-cofilin-1 (27 cells). Statistical significance was determined by one-way ANOVA and a Dunnet's post hoc test. All data are representative of at least three independent experiments. *p ≤ 0.05, **p ≤ 0.01, ***p ≤ 0.001, and ****p ≤ 0.0001.

The online version of this article includes the following figure supplement(s) for figure 3:

**Figure supplement 1.** Cytochalasin B blocks access to cortical barbed ends.

## Cofilin-1 supports both actin turnover and myosin contractility at leader bleb necks

Because the cortical actin flow in leader blebs is driven by myosin, we wondered if ADF and/or cofilin-1 are important for cortical contractility. To address this possibility, we determined the compressibility of cells using a previously described gel sandwich approach (*Liu et al., 2015*). Briefly, freshly

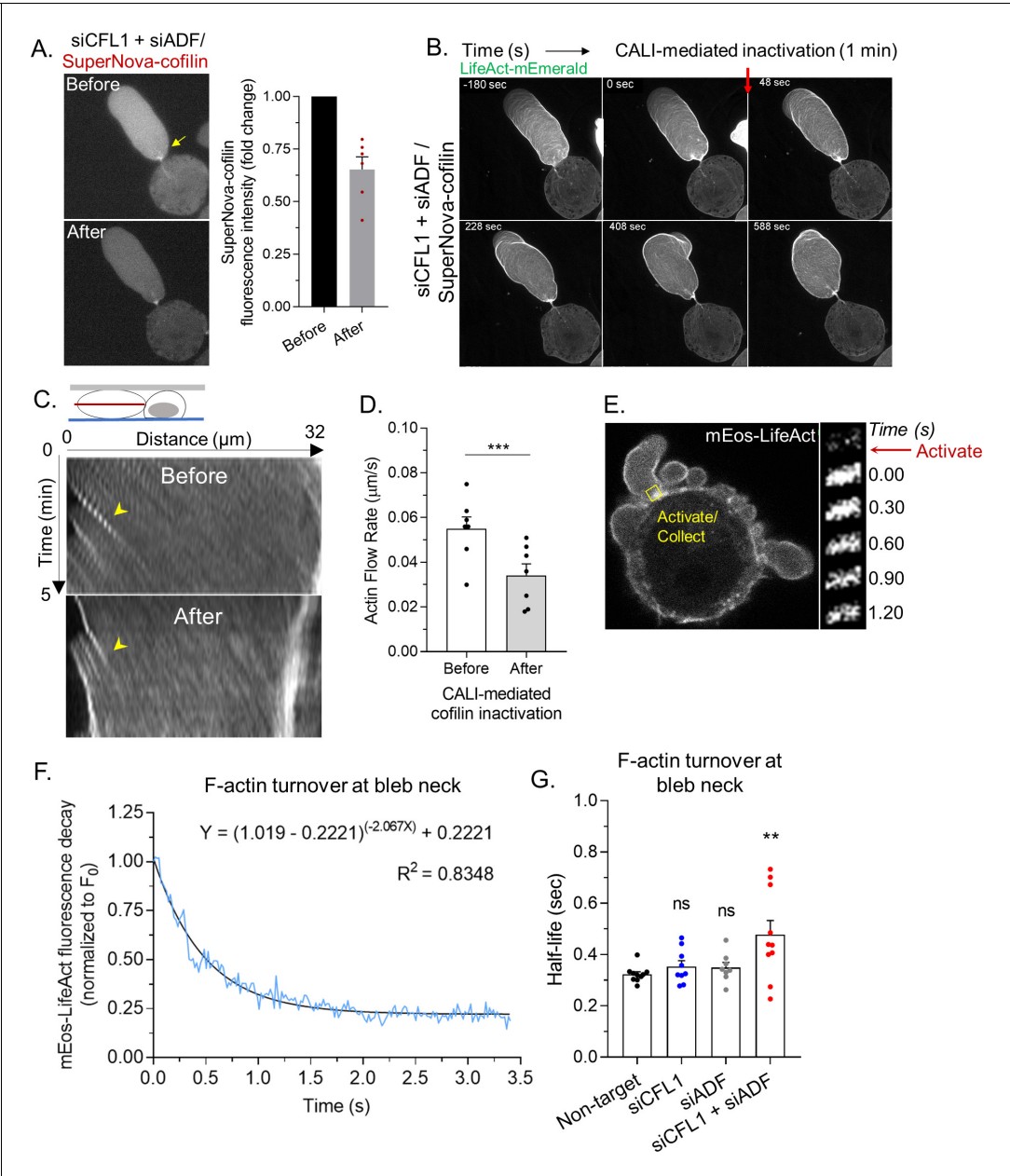

**Figure 4.** Rapid cortical actin flow requires actin depolymerizing factor (ADF) and cofilin-1 severing at leader bleb necks. (**A**) *Left*, SuperNova-cofilin-1 localization in cells depleted of endogenous cofilin-1 and ADF by RNAi before and after 1 min of red light irradiation. *Right*, quantitative analysis of CALI, as determined by the fold change in SuperNova emission. (**B**) Montage of mEmerald-LifeAct before and after cofilin-1 inactivation in a cell depleted of endogenous cofilin-1 and ADF by RNAi. (**C**) Kymographs of cortical actin (mEmerald-LifeAct) flow from the leader bleb tip before and after cofilin-1 inactivation. (**D**) Quantitative evaluation of cortical actin flow rates before and after cofilin-1 inactivation. Statistical significance was determined by a paired Student's t-test. (**E**) *Left*, representative image of a freshly plated (spherical) cells with mEos3.2-LifeAct. *Right*, montage of mEos3.2-LifeAct within the shown ROI before and after photoactivation. (**F**) Average decay curve for mEos3.2-LifeAct at bleb necks (normalized to the initial fluorescence level; $F/F_0$). The curve was fit using a non-linear single phase decay function. (**G**) $t_{1/2}$ for mEos3.2-LifeAct after photoactivation at bleb necks for non-targeting, CFL1, ADF, and CFL1 + ADF RNAi. Statistical significance was determined by one-way ANOVA and a Dunnet's post hoc test. All data are representative of at least three independent experiments. *$p \leq 0.05$, **$p \leq 0.01$, ***$p \leq 0.001$, and ****$p \leq 0.0001$.

The online version of this article includes the following figure supplement(s) for figure 4:

**Figure supplement 1.** Actin turnover measurement controls.

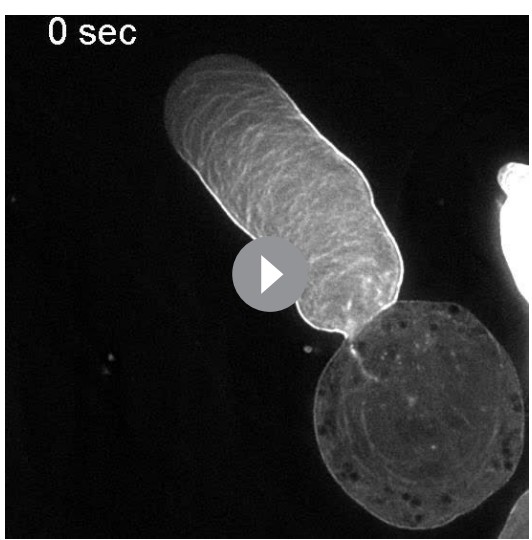

**Video 7.** Time-lapse imaging of a melanoma A375-M2 cell confined down to 3 μm with mEmerald-LifeAct after chromophore assisted light inactivation (CALI) of SuperNova-cofilin-1. The cell was depleted of endogenous actin depolymerizing factor (ADF) and cofilin-1 by RNAi.
https://elifesciences.org/articles/67856#video7

trypsinized (spherical) cells are placed between two polyacrylamide gels of known stiffness (1 kPa). Subsequently, the ratio of the cell height (*h*) to the diameter (*d*) is used to determine compressibility (*Figure 5A*). After depleting cofilin-1, we found cells to be significantly more compressible, whereas depleting ADF had no effect (*Figure 5A*). Moreover, in cofilin-1 depleted cells, the depletion of ADF did not have any additional effect on cell compressibility (*Figure 5A*). Reconstituting ADF and cofilin-1 RNAi cells with RFP-cofilin-1 rescued cell compressibility to levels similar to control, whereas RFP-ADF was unable to rescue compressibility (*Figure 5B–C*). Therefore, cofilin-1 may be particularly important for contractility. In support of this idea, cofilin-1 has been shown to modulate contractility through a variety of mechanisms, which involve changes in both actin turnover and network architecture (*Chan et al., 2009*; *Chugh et al., 2017*; *Ennomani et al., 2016*). As determined by immunofluorescence imaging of phosphorylated regulatory light chain (pRLC), we could confirm that signaling to myosin was not significantly affected by depleting ADF and/or cofilin-1 (*Figure 5D–E*). In confined cells, depleting ADF and cofilin-1 leads to the elongation of leader bleb necks, which are decorated with myosin (EGFP-RLC; *Figure 5F*). We then used CALI to determine if the accumulation of myosin at leader blebs necks is directly caused by the removal of these proteins. Indeed, within minutes of SuperNova-cofilin-1 inactivation, we observed myosin accumulating at leader bleb necks (EGFP-RLC; *Figure 5G* and *Video 8*). Additionally, the flow of myosin toward leader bleb necks was significantly impeded after cofilin-1 inactivation, as determined by tracking individual myosin minifilaments (*Figure 5H*). While ADF may be particularly important for actin turnover, these results point to cofilin-1 as having important roles in both actin turnover and myosin contractility at leader bleb necks.

## Discussion

Here, we identify ADF and cofilin-1 as key mediators of the rapid (cortical) actin flow in leader blebs. This is significant as the rapid flow of cortical actin in leader blebs is essential for confined migration (*Liu et al., 2015*; *Ruprecht et al., 2015*; *Bergert et al., 2015*). We report that melanoma cells depleted of cofilin-1 poorly undergo LBBM, whereas removing ADF did not have a significant effect. Because ADF and cofilin-1 are thought to have redundant or overlapping roles, we were surprised to then find that depleting cofilin-1 together with ADF led to a near complete inhibition of LBBM. Therefore, we set out

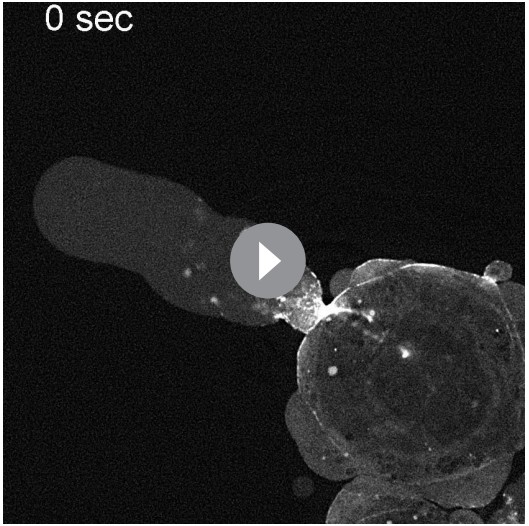

**Video 8.** Time-lapse imaging of a melanoma A375-M2 cell confined down to 3 μm with EGFP tagged regulatory light chain (EGFP-RLC) after chromophore assisted light inactivation (CALI) of SuperNova-cofilin-1. The cell was depleted of endogenous actin depolymerizing factor (ADF) and cofilin-1 by RNAi.
https://elifesciences.org/articles/67856#video8

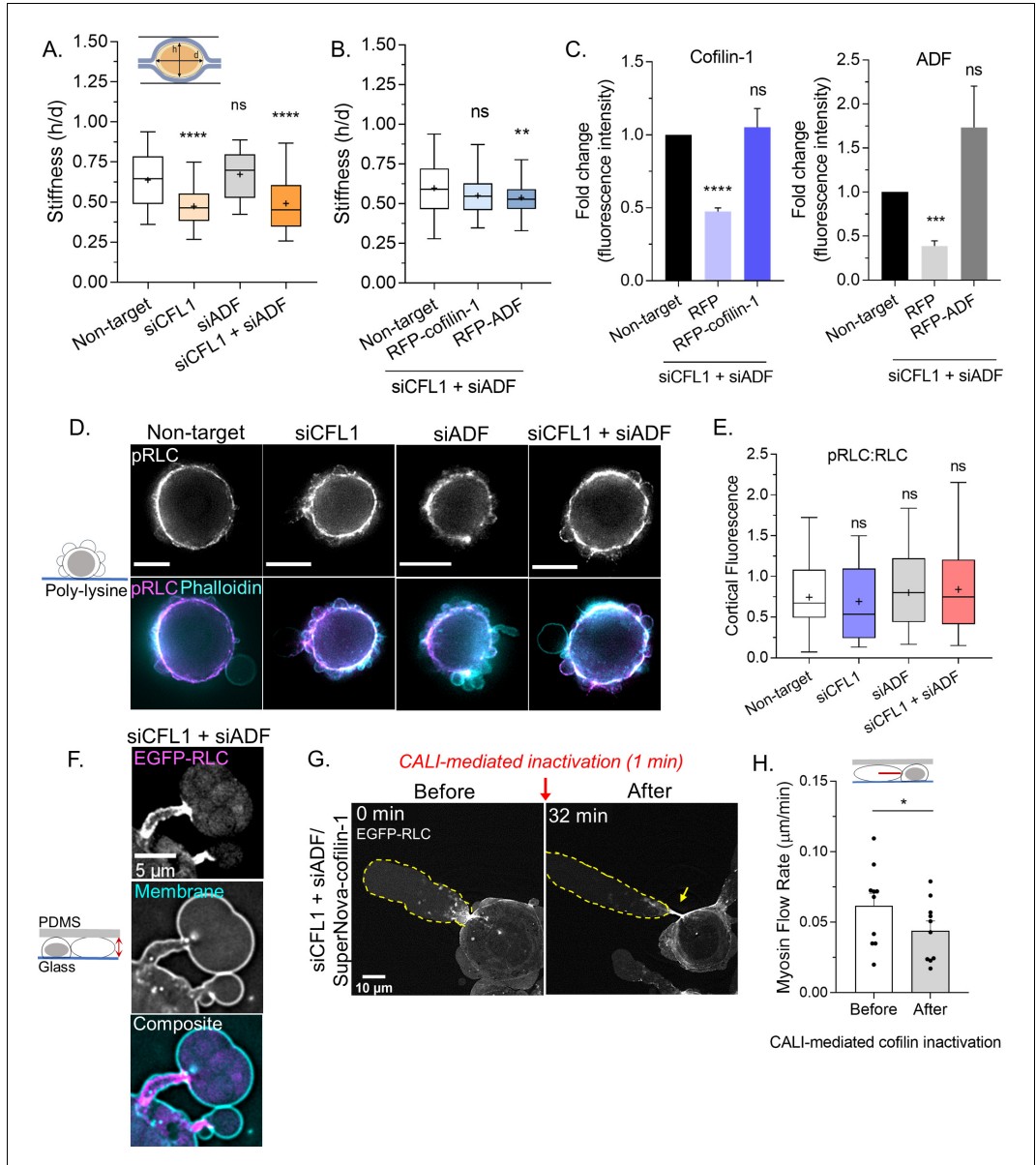

**Figure 5.** Cofilin-1 supports both actin turnover and myosin contractility at leader bleb necks. (A) A previously described gel sandwich assay was used to measure the stiffness (*h/d*) of spherical cells after non-targeting (91 cells), CFL1 (30 cells), actin depolymerizing factor (ADF) (25 cells), and CFL1 + ADF RNAi (42 cells). (B) Cell stiffness (*h/d*) of spherical cells with non-targeting (181 cells) or siCFL1 + siADF RNAi rescued with RFP-cofilin-1 (41 cells) or RFP-ADF (74 cells). (A–B) Statistical significance was determined by one-way ANOVA and a Dunnet's post hoc test. (C) Cofilin-1 (*left*) and ADF (*right*) levels (fluorescence intensity; fold change) of adhered RNAi cells by immunofluorescence confirming rescue by transfection with RFP-cofilin-1 (*left*) or RFP-ADF (*right*). Statistical significance was determined by an unpaired one-sample t-test. (D) Immunofluorescence imaging of endogenous phosphorylated regulatory light chain (pRLC) (S18), total RLC, and filamentous-actin (F-actin) (phalloidin) in freshly plated (spherical) cells after non-targeting, CFL1, and CFL1 + ADF RNAi. (E) Ratio of cortical pRLC (S18) to total RLC fluorescence intensity after non-targeting (114 cells), CFL1 (107 cells), ADF (124 cells), and CFL1 + ADF RNAi (91 cells). Statistical significance was determined by one-way ANOVA and a Dunnet's post hoc test. (F) Localization of EGFP tagged regulatory light chain (EGFP-RLC) in a cell confined down to 3 μm after CFL1 + ADF RNAi. (G) EGFP-RLC dynamics in a cell depleted of cofilin-1 and ADF before and after chromophore assisted light inactivation (CALI) of SuperNova-cofilin-1. Arrow points to myosin accumulating at an elongating leader bleb neck after cofilin-1 inactivation. (H) Myosin minifilament flow rate before and after cofilin-1 inactivation. Statistical significance was determined using a paired Student's t-test. All data are representative of at least three independent experiments. *p ≤ 0.05, **p ≤ 0.01, ***p ≤ 0.001, and ****p ≤ 0.0001.

The online version of this article includes the following source data for figure 5:

**Source data 1.** Raw stiffness measurements.

to determine the basis for this effect in cancer cells.

Initially, we used high spatial and temporal resolution imaging for close inspection of RNAi cells. Cells lacking cofilin-1 alone had much smaller blebs. Without both cofilin-1 and ADF, many cells had blebs with elongated necks that would not retract. A similar elongated neck phenotype was observed by Wiggan et al., whereas they reported an increase in the frequency of HeLa cells with blebs after cofilin-1 RNAi (*Wiggan et al., 2012*). While these authors contend that cofilin-1 inhibits contractility by competing with myosin for F-actin binding, our results are in line with work showing that cofilin-1 increases contractility through optimizing actin filament lengths and de-branching (*Chan et al., 2009*; *Chugh et al., 2017*; *Ennomani et al., 2016*). This led us to wonder if ADF and/or cofilin-1 may be important for actin turnover at bleb necks. In agreement with this concept, in cells lacking endogenous cofilin-1 and ADF, EGFP-cofilin-1 was enriched at bleb necks.

In confined cells, we observed a dramatic accumulation of actin at bleb necks. In order to better understand how ADF and cofilin-1 regulate the overall level of cortical actin, we utilized spherical cells which predominantly have cortical actin. Using flow cytometry, we found that depleting cells of cofilin-1 could significantly increase the overall level of actin. Although ADF on its own had no effect, depleting cofilin-1 with ADF increased the overall level of actin even further, which might suggest that ADF augments the severing activity of cofilin-1. Similarly, the level of cortical barbed ends was significantly increased in the absence of cofilin-1, whereas removing ADF marginally increased the level of cortical barbed ends. Depleting both cofilin-1 and ADF appeared to have an additive effect, leading to the largest increase in the level of uncapped (polymerization competent) barbed ends. This result suggests that cofilin-1 severing, in collaboration with ADF, leads to the rapid disassembly of cortical actin and not polymerization at new barbed ends.

Because changes in the cellular F/G-actin ratio can trigger specific transcriptional programs, such as through the activation of the transcription factor, MRTF-A, we wanted to determine if the effects we observed on LBBM are a direct result of down-regulating actin severing (*Olson and Nordheim, 2010*). For this, we utilized CALI for the rapid inactivation of SuperNova-cofilin-1. In cells depleted of ADF and cofilin-1, SuperNova-coflin-1 was sufficient to restore normal bleb morphology and dynamics. However, within minutes of cofilin-1 inactivation, actin began to accumulate at elongated bleb necks. By kymograph analysis, we also observed a significant decrease in the cortical actin flow rate. Using a photoactivatable LifeAct construct, we then directly measured the rate of actin turnover (i.e., fluorescence decay) at bleb necks. While depleting ADF or cofilin-1 alone did not have a significant effect on actin turnover rates, depleting both proteins led to a large increase in the rate of actin turnover at bleb necks. Thus, in cells lacking ADF and cofilin-1, defects in bleb morphology and dynamics correlate with a reduction in actin turnover at the neck.

Cofilin-1 has been previously implicated in regulating actomyosin contractility. Therefore, using spherical cells, we determined the overall effect of ADF and cofilin-1 on cortical contractility. As indicated by increased compressibility, cells were found to be less stiff after depleting cofilin-1. In contrast, removing ADF had little effect on cell stiffness. Moreover, depleting both ADF and cofilin-1 was similar to removing cofilin-1 alone and could be rescued by an RNAi resistant version of cofilin-1 but not ADF. Therefore, cofilin-1 may be particularly important to support actomyosin contractility. In line with this notion, cofilin-1 has been shown to support myosin contractility through optimizing actin filament lengths and de-branching (*Chan et al., 2009*; *Chugh et al., 2017*; *Ennomani et al., 2016*). As indicated by the lack of any change in pRLC levels or localization, removing these proteins does not appear to effect signaling to myosin. The role of cofilin-1 in supporting myosin contractility is further supported by CALI. More specifically, we demonstrate that inactivating cofilin-1 decreases the rate at which myosin minifilaments flow toward the bleb neck.

Thus, our data are consistent with a model whereby ADF and cofilin-1 play key roles during LBBM. More specifically, we assert that ADF and cofilin-1, together, optimize actin disassembly and myosin contractility at bleb necks (*Figure 6*, *top*). Whereas, in the absence of these proteins, incoming cortical actin fails to disassemble and accumulates with myosin at the elongated necks of persistent blebs (*Figure 6*, *below*). Collectively, this study further points to rapid (cortical) actin flows as being essential for confined (leader bleb-based) migration. This is significant as many cancer cells have been shown to undergo LBBM (*Logue et al., 2015*; *Liu et al., 2015*).

This work also reveals an unanticipated role for ADF. In melanoma cells, ADF appears to augment the activity of cofilin-1 at bleb necks. Largely, ADF and cofilin-1 are thought to have redundant or overlapping roles (*Hotulainen et al., 2005*). However, ADF has been shown to have significant

monomer sequestering activity (*Chen et al., 2004*). We speculate that, in the absence of both proteins, dampened actin severing coupled with uncontrolled filament elongation contributes to the severe phenotypes we observe. Thus, the dissemination of melanoma tumors is likely to be blocked by the simultaneous inhibition of ADF and cofilin-1. As actin severing by ADF and cofilin-1 can be regulated by a number of functional interactions, such as with Aip1 and cyclase-associated protein 1, future work will endeavor to determine the contribution of these factors in regulating the cortical actomyosin flow in fast amoeboid cells (*Bertling et al., 2004*; *Nadkarni and Brieher, 2014*; *Chen et al., 2015*; *Shekhar et al., 2019*; *Kotila et al., 2019*).

# Materials and methods

## Key resources table

| Reagent type (species) or resource | Designation | Source or reference | Identifiers | Additional information |
|---|---|---|---|---|
| Cell line (*Homo sapiens*) | A375-M2 | ATCC | CRL-3223 | Metastatic melanoma |
| Cell line (*Homo sapiens*) | A549 | ATCC | CCL-185 | Lung adenocarcinoma |
| Chemical compound, drug | SYLGARD 184 | Dow Corning | Cat no. 24236–10 | PDMS |
| Transfected construct (*Homo sapiens*) | EGFP-cofilin-1 WT, S3A, and S3E | Addgene (a gift from Dr James Bamburg) | Plasmid no. 50859, 50854, and 50855 | Plasmid constructs to transfect |
| Transfected construct (*Homo sapiens*) | RFP-cofilin-1 | Dr James Bamburg (Colorado State University) | n/a | Plasmid construct to transfect |
| Transfected construct (*Homo sapiens*) | RFP-ADF | Dr James Bamburg (Colorado State University) | n/a | Plasmid construct to transfect |
| Transfected construct (*Homo sapiens*) | SuperNova-cofilin-1 | Dr Kazuyo Sakai (Osaka University, Osaka, Japan) | n/a | Plasmid construct to transfect and destroy cofilin-1 by CALI |
| Transfected construct (*Saccharomyces cerevisiae*) | mEos3.2-LifeAct | Addgene (a gift from Michael Davidson) | Plasmid no. 54696 | Plasmid construct to transfect and monitor F-actin dynamics |
| Sequence-based reagent (*Homo sapiens*) | Non-targeting siRNA | Thermo Fisher | Cat no. 4390844 | Control siRNA to transfect |
| Sequence-based reagent (*Homo sapiens*) | Cofilin-1 siRNA | Thermo Fisher | Cat no. 4392420; s2936 | Cofilin-1 siRNA to transfect |
| Sequence-based reagent (*Homo sapiens*) | ADF siRNA | Thermo Fisher | Cat no. 4392422; s21737 | ADF siRNA to transfect |
| Antibody | Anti-cofilin-1 (mouse monoclonal) | Thermo Fisher | Cat no. MA5-17275 | WB (1:1000), IF (1:250) |
| Antibody | Anti-ADF (mouse monoclonal) | Thermo Fisher | Cat no. MA5-25485 | WB (1:1000), IF (1:250) |
| Sequence-based reagent (*Homo sapiens*) | Cofilin-1 forward qPCR primer | Thermo Fisher | n/a | GCAACCTATGAGA CCAAGGAGAG |

*Continued on next page*

*Continued*

| Reagent type (species) or resource | Designation | Source or reference | Identifiers | Additional information |
|---|---|---|---|---|
| Sequence-based reagent (*Homo sapiens*) | ADF forward qPCR primer | Thermo Fisher | n/a | GCACCAGAACTA GCACCTCTGA |
| Sequence-based reagent (*Homo sapiens*) | GAPDH forward qPCR primer | Thermo Fisher | n/a | GTCTCCTCTGACT TCAACAGCG |
| Recombinant DNA protein | Alexa Fluor 568-conjugated G-actin from rabbit muscle | Thermo Fisher | Cat no. A12374 | Fluorescent G-actin to label actin barbed ends |
| Software | Fiji | n/a | https://imagej.net/Fiji | Microscopy |
| Software | Prism | GraphPad | n/a | Statistical analyses |
| Software | BioRender | Toronto, ON | n/a | Illustration |
| Other | DeltaVision Elite | GE | n/a | Commercial deconvolution microscopy system |
| Other | LSM880 with fast Airy Scan | Zeiss | n/a | Commercial point scanning confocal microscopy system |

## Cell culture

A375-M2 (CRL-3223) and A549 (CCL-185) were obtained from the American Type Culture Collection (ATCC; Manassas, VA). Cells were cultured in high-glucose DMEM supplemented with 10% FBS (cat no. 12106C; Sigma Aldrich, St. Louis, MO), GlutaMAX (Thermo Fisher, Carlsbad, CA), antibiotic-antimycotic (Thermo Fisher), and 20 mM HEPES at pH 7.4 for up to 30 passages. Cells were tested, and negative, for mycoplasma using MycoAlert PLUS Mycoplasma Detection Kit (Lonza, Walkersville, MD).

## Confinement

This protocol has been described in detail elsewhere (*Logue et al., 2018*). Briefly, PDMS (cat no. 24236–10; Dow Corning 184 SYLGARD) was purchased from Krayden (Westminster, CO); 2 mL was cured overnight at 37°C in each well of a six-well glass bottom plate (cat no. P06-1.5H-N; Cellvis, Mountain View, CA). Using a biopsy punch (cat no. 504535; World Precision Instruments, Sarasota, FL), an 8 mm hole was cut and 3 mL of serum-free media containing 1% BSA was added to each well and incubated overnight at 37°C. After removing the serum-free media containing 1% BSA, 300 µL of complete media containing trypsinized cells (250,000 to 1 million) and 2 µL of 3.11 µm beads (cat no. PS05002; Bangs Laboratories, Fishers, IN) were then pipetted into the round opening. The vacuum created by briefly lifting one side of the hole with a 1 mL pipette tip was used to move cells and beads underneath the PDMS. Finally, 3 mL of complete media was added to each well and cells were recovered for ~60 min before imaging.

## Plasmids

SuperNova-cofilin-1 was a gift from Dr Kazuyo Sakai (Osaka University, Osaka, Japan). EGFP-cofilin-1 WT (no. 50859; a gift from Dr James Bamburg), S3A (no. 50854; a gift from Dr. James Bamburg), S3E (no. 50855; a gift from Dr James Bamburg), and mEos3.2-LifeAct (no. 54696; a gift from Michael Davidson) were obtained from Addgene (Watertown, MA). RFP-cofilin-1 and RFP-ADF were a gift from Dr James Bamburg (Colorado State University); 1 µg of plasmid was used to transfect 400,000 cells in each well of a six-well plate using Lipofectamine 2000 (5 µL; Thermo Fisher) in OptiMEM (400 µL; Thermo Fisher). After 20 min at room temperature, plasmid in Lipofectamine 2000/OptiMEM was then incubated with cells in complete media (2 mL) overnight.

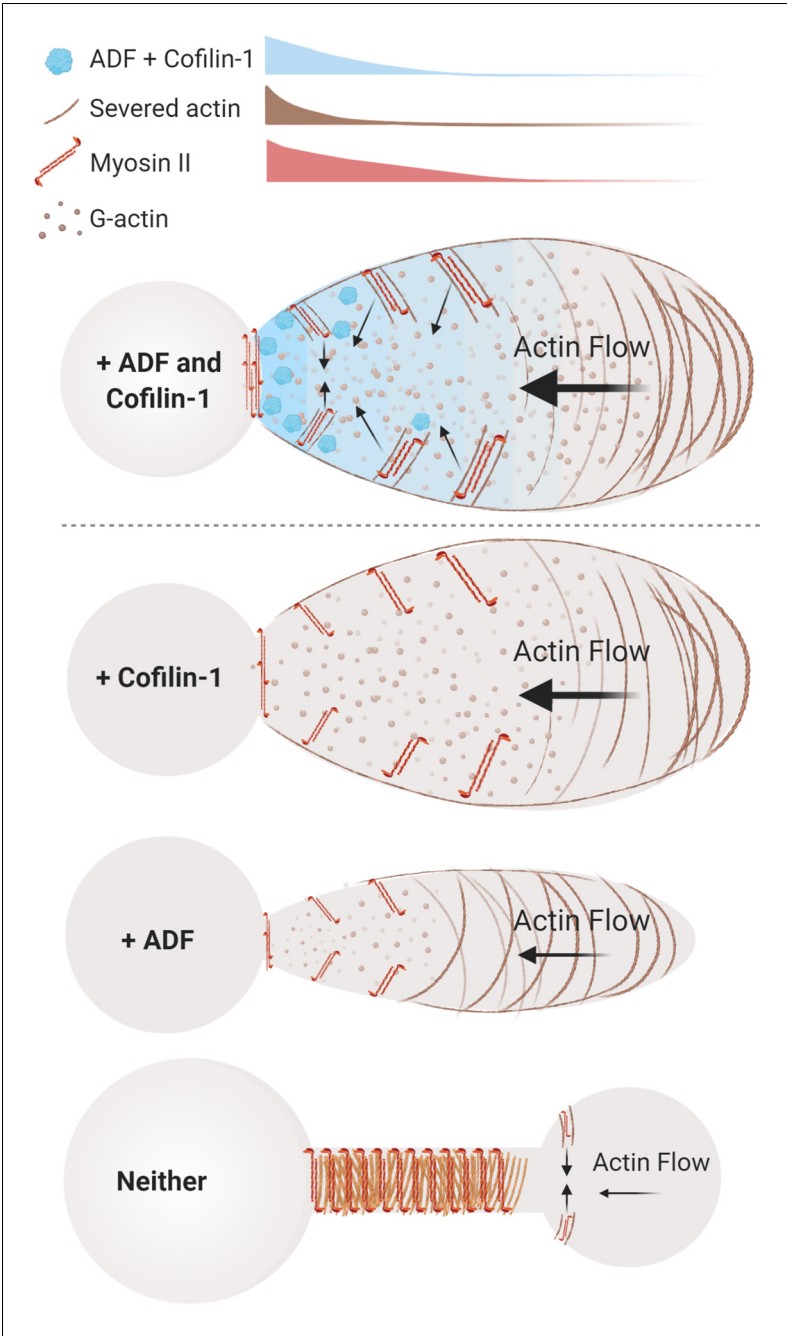

**Figure 6.** Model of actin depolymerizing factor (ADF) and cofilin-1 function within leader blebs. *Top*, in the presence of both ADF and cofilin-1, cells display large blebs with rapid cortical actin flow. *Below*, in the absence of cofilin-1 or ADF, cells form smaller blebs with slower cortical actin flow. *Bottom*, without both ADF and cofilin-1, blebs display several defects, including a failure to retract and an accumulation of actomyosin at elongated necks.

## Pharmacological treatments

Latrunculin-A (cat no. 3973) and cytochalasin B (cat no. 5474) were purchased from Tocris Bioscience (Bristol, UK). DMSO (Sigma Aldrich) was used to make 5 and 2 mM stock solutions of Latrunculin-A and cytochalasin B, respectively. To disassemble actin, cells resuspended in flow buffer were treated with 5 μM Latrunculin-A for 10 min at room temperature before flow cytometry. For barbed end assays, cytochalasin B was pre-diluted in complete media before it was incubated with cells for 1 hr at 37°C.

## Locked nucleic acids

Non-targeting (cat no. 4390844), cofilin-1 (cat no. 4392420; s2936), and ADF (cat no. 4392422; s21737) locked nucleic acids (LNAs) were purchased from Thermo Fisher. All LNA transfections were performed using RNAiMAX (5 μL; Thermo Fisher) and OptiMEM (400 μL; Thermo Fisher); 100,000 cells were trypsinized and seeded in six-well plates in complete media. After cells adhered (~1 hr), LNAs in RNAiMAX/OptiMEM were added to cells in complete media (2 mL) at a final concentration of 50 nM. Cells were incubated with LNAs for 2 days.

## Western blotting

Whole-cell lysates were prepared by scraping cells into ice-cold RIPA buffer (50 mM HEPES pH 7.4, 150 mM NaCl, 5 mM EDTA, 0.1% SDS, 0.5% deoxycholate, and 1% Triton X-100) containing protease and phosphatase inhibitors (Roche, Switzerland). Before loading onto 4–12% NuPAGE Bis-Tris gradient gels (Thermo Fisher), DNA was sheared by sonication and samples were boiled for 10 min in loading buffer. Following SDS-PAGE, proteins in gels were transferred to nitrocellulose membranes and subsequently immobilized by air drying overnight. After blocking in Tris-buffered saline containing 0.1% Tween 20 and 1% BSA, primary antibodies against cofilin-1 (cat no. MA5-17275; Thermo Fisher) and ADF (cat no. MA5-25485; Thermo Fisher and cat no. D8818; Sigma Aldrich) were incubated with membranes overnight at 4°C. Bands were then resolved with IRDye conjugated secondary antibodies on an Odyssey scanner from LI-COR Biosciences, Lincoln, NE. GAPDH (cat no. 97166; Cell Signaling Technology, Danvers, MA) was used to confirm equal loading.

## qRT-PCR

Total RNA was isolated from cells using the PureLink RNA Mini Kit (Thermo Fisher) and was used for reverse transcription using a high-capacity cDNA Reverse Transcription Kit (Applied Biosystems, Foster City, CA). cDNA was used for qRT-PCR using PowerUp SYBR Green Master Mix (Thermo Fisher) and primers, human CFL1: GCAACCTATGAGACCAAGGAGAG (forward sequence), human ADF: GCACCAGAACTAGCACCTCTGA (forward sequence), and human GAPDH: GTCTCCTCTGACTTCAACAGCG (forward sequence), on a CFX96 real-time PCR detection system (Bio-Rad). Relative mRNA levels were calculated by the ΔCt method.

## Transmigration

Transmigration assays were performed using polycarbonate filters with 8 or 12 μm pores (Corning; Corning, NY). Prior to the assays, cells were serum-starved for 24 hr and polycarbonate filters were fibronectin (10 μg/mL; Millipore, Burlington, MA) coated for 1 hr followed by air drying; 100,000 cells in serum-free media were seeded in the top chamber while the bottom chamber contained media with 20% FBS to attract cells. After 24 hr, cells from the bottom of the filter were trypsinized and counted using an automated cell counter (TC20; Bio-Rad, Hercules, CA). Transmigration was then calculated as the ratio of cells on the bottom of the filter vs. the total.

## Flow cytometry

Roughly $1 \times 10^6$ trypsinized cells in flow buffer (HBS with 1% BSA) were fixed using 4% paraformaldehyde (cat no. 15710; Electron Microscopy Sciences, Hatfield, PA) for 20 min at room temperature. After washing, cell pellets were resuspended in flow buffer and incubated with 0.1% Triton X-100, Alexa Fluor 647-conjugated phalloidin (cat no. A22287; Thermo Fisher), and DAPI (Sigma Aldrich) for 30 min at room temperature. Data were acquired on a FACSCalibur (BD Biosciences, Franklin Lakes, NJ) flow cytometer. Flow cytometric analysis was performed using FlowJo (Ashland, OR) software.

## Barbed end assay

The protocol for the barbed end assay was used with minor modifications (*Vitriol et al., 2007*; *Symons and Mitchison, 1991*). Prior to barbed end assays, cells were trypsinized and plated on poly-L-lysine coated six-well glass bottom plates (Cellvis). To allow for a minimal level of cell attachment, cells were incubated for 10 min in a tissue culture incubator. Cells were then gently permeabilized for 1 min with saponin buffer (138 mM KCl, 4 mM MgCl$_2$, 3 mM EGTA, 0.1% saponin, 1 mM ATP, 3 μM phalloidin, and 1% BSA) followed by one wash with saponin-free buffer. Permeabilized

cells were then incubated with Alexa Fluor 568-conjugated G-actin from rabbit muscle (cat no. A12374; Thermo Fisher) for 3 min in a tissue culture incubator and washed with saponin-free buffer. Treated cells were then fixed with 4% paraformaldehyde in HEPES-buffered saline (HBS), washed with HBS alone, and immediately imaged.

## Immunofluorescence

After washing with HBS, cells in six-well glass bottom plates (Cellvis) were fixed with 4% paraformaldehyde (Electron Microscopy Sciences) in HBS for 20 min at room temperature. Blocking, permeabilization, antibody incubation, and washing were done in HBS with 1% BSA, 1% fish gelatin, 0.1% Triton X-100, and 5 mM EDTA. A 1:250 dilution of ADF (cat no. MA5-25485; Thermo Fisher), cofilin-1 (cat no. MA5-17275; Thermo Fisher), pRLC (cat no. PA5-17727 or MA5-15163; Thermo Fisher), or RLC (cat no. PA5-17624; Thermo Fisher) antibody was incubated with cells overnight at 4°C. After extensive washing, a 1:400 dilution of Alexa Fluor 488-conjugated anti-rabbit secondary antibody (cat no. A-21206; Thermo Fisher) was then incubated with cells for 2 hr at room temperature. Cells were then incubated with a 1:250 dilution of Alexa Fluor 568-conjugated phalloidin (cat no. A12380; Thermo Fisher) and a 1:1000 dilution of DAPI (cat no. D1306; Thermo Fisher). Cells were again extensively washed and then imaged in HBS. Fluorescence intensity of cofilin-1 or ADF was measured in adhered cells with either EGFP-cofilin-1, RFP-cofilin-1, RFP-ADF, or an empty vector fluorescent protein. A minimum of 26 cells over three independent experiments were averaged and normalized to the control in each experimental $n$-value.

## Cell stiffness assay

The protocol for the gel sandwich assay was used with minor modifications (*Liu et al., 2015*). Six-well glass bottom plates (Cellvis) and 18 mm coverslips were activated using 3-aminopropyltrimethoxysilane (Sigma Aldrich) for 5 min and then for 30 min with 0.5% glutaraldehyde (Electron Microscopy Sciences) in PBS; 1 kPa polyacrylamide gels were made using 2 µL of blue fluorescent beads (200 nm; Thermo Fisher), 18.8 µL of 40% acrylamide solution (cat no. 161–0140; Bio-Rad), and 12.5 µL of bis-acrylamide (cat no. 161–0142; Bio-Rad) in 250 µL of PBS. Finally, 2.5 µL of ammonium persulfate (10% in water) and 0.5 µL of tetramethylethylenediamine was added before spreading 9 µL drops onto treated glass under coverslips. After polymerizing for 40 min, the coverslip was lifted in PBS, extensively rinsed and incubated overnight in PBS. Before each experiment, the gel attached to the coverslip was placed on a 14 mm diameter, 2 cm high PDMS column for applying a slight pressure to the coverslip with its own weight. Then, both gels were incubated for 30 min in media before plates were seeded. After the bottom gels in plates was placed on the microscope stage, the PDMS column with the top gel was placed on top of the cells seeded on the bottom gels, confining cells between the two gels. After 1 hr of adaptation, the height of cells was determined with beads by measuring the distance between gels, whereas the cell diameter was measured using a far-red PM dye (cat no. C10046; Thermo Fisher). Stiffness was defined as the height ($h$) divided by the diameter ($d$).

## Microscopy

Live high-resolution imaging was performed using a General Electric (Boston, MA) DeltaVision Elite imaging system mounted on an Olympus (Japan) IX71 stand with a computerized stage, environment chamber (heat, $CO_2$, and humidifier), ultrafast solid-state illumination with excitation/emission filter sets for DAPI, CFP, GFP, YFP, and Cy5, critical illumination, Olympus PlanApo N 60X/1.42 NA DIC (oil) objective, Photometrics (Tucson, AZ) CoolSNAP HQ2 camera, proprietary constrained iterative deconvolution, and vibration isolation table.

## Chromophore assisted light inactivation

Live confined cells co-transfected with SuperNova-cofilin-1 and either mEmerald-LifeAct or EGFP-RLC were imaged every 12 s for 5 min before CALI. Cells were then subjected to 1 min of high intensity red light irradiation (DeltaVision Elite). Images were subsequently acquired every 12 s for 1 hr post-irradiation.

## Actomyosin flow rates

Actin and myosin minifilament flow rates were calculated from images taken every 12 s of mEmerald-LifeAct and EGFP-RLC, respectively. Kymographs of each were generated in Fiji (https://imagej.net/Fiji) and spanned the length of leader blebs.

## Fluorescence loss after photobleaching

Cells transiently transfected with the photoactivatable construct, mEos3.2-LifeAct, were imaged using a Zeiss (Germany) laser scanning confocal microscope (LSM880) with fast Airy Scan. Regions at bleb necks were photo-converted using the 405 nm laser at 20% power (three iterations). Regions were sampled every 15 ms for 10 and 150 cycles before and after photo-conversion, respectively. Fluorescence measurements were acquired using ZEN software (Zeiss) and decay was calculated as $F/F_0$. Non-linear one-phase decay curves were fit to data using Prism (GraphPad, San Diego, CA).

## Cell migration

To perform cell speed, plot of origin, and direction autocorrelation analyses, we used an Excel (Microsoft, Redmond, WA) plugin, DiPer, developed by Gorelik and colleagues and the Fiji plugin, MTrackJ, developed by Erik Meijering for manual tracking (*Gorelik and Gautreau, 2014*; *Meijering et al., 2012*). For minimizing positional error, cells were tracked every other frame. Bright-field imaging was used to confirm that beads were not obstructing the path of a cell. Cells that traveled a distance equivalent to at least one cell length over the course of the 5 hr time-lapse were classified as highly motile.

## Bleb morphology and dynamics

For leader bleb and total bleb areas, freshly confined cells were traced from high-resolution images with the free-hand circle tool in Fiji (https://imagej.net/Fiji). From every other frame, the percentage of cell area for leader blebs and percentage of cell area for total blebs were calculated in Excel (Microsoft). Frame-by-frame measurements were then used to generate an average for each cell. Bleb retraction rates were determined by dividing the bleb length by the amount of time taken to completely retract the bleb into the cell body. For each cell, retraction rates were calculated from two to three blebs.

## Cofilin-1 enrichment

For regional analyses of EGFP-cofilin-1 distribution, 10 ROIs each representing 10% of the length of the leader bleb were drawn using Fiji (https://imagej.net/Fiji). Mean gray values of each ROI were normalized to the first ROI taken at the bleb neck.

## Statistics

All box plots are Tukey in which '+' and line denote the mean and median, respectively. Sample sizes were determined empirically and based on saturation. As noted in each figure legend, statistical significance was determined by either a two-tailed Student's t-test or multiple-comparison test post hoc. Normality was determined by a D'Agostino and Pearson test in Prism (GraphPad). *$p \leq$ 0.05, **$p \leq$ 0.01, ***$p \leq$ 0.001, and ****$p \leq$ 0.0001.

## Illustration

The model for ADF and cofilin-1 function was drawn in BioRender (Toronto, ON).

## Acknowledgements

We thank members of the Logue Lab for insightful discussions and especially, Dr Sandrine B Lavenus, for help with cell stiffness measurements. We would also like to thank the administrative staff within the Department of Regenerative and Cancer Cell Biology at the Albany Medical College. This work was supported by a Young Investigator Award from the Melanoma Research Alliance (MRA; award no. 688232) and a Research Scholar Grant from the American Cancer Society (ACS; award no. RSG-20-019-01 - CCG).

## Additional information

### Funding

| Funder | Grant reference number | Author |
|---|---|---|
| Melanoma Research Alliance | 688232 | Maria F Ullo<br>Jeremy S Logue |
| American Cancer Society | RSG-20-019-01 - CCG | Jeremy S Logue |

The funders had no role in study design, data collection and interpretation, or the decision to submit the work for publication.

### Author contributions

Maria F Ullo, Investigation, Writing - original draft; Jeremy S Logue, Conceptualization, Supervision, Funding acquisition, Writing - review and editing

### Author ORCIDs

Jeremy S Logue  https://orcid.org/0000-0002-5274-2052

### Decision letter and Author response

Decision letter https://doi.org/10.7554/eLife.67856.sa1
Author response https://doi.org/10.7554/eLife.67856.sa2

## Additional files

### Supplementary files

• Transparent reporting form

### Data availability

Source data files for Figures 1, 2, and 5 have been provided.

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
