## [Decision Letter]

[Editors' note: this paper was reviewed by Review Commons.]

**Acceptance summary:**

Your work clarifies the contribution of two isoforms of the ADF/cofilin family of proteins for bleb-based migration. It also highlights beautifully one more time how crucial the precise control of actin turnover is for many actin-based cellular processes.

**Decision letter after peer review:**

Thank you for submitting your article "ADF and cofilin-1 collaborate to promote cortical actin flow and the leader bleb-based migration of confined cells" for consideration by *eLife*. Your article has been reviewed by 2 peer reviewers at Review Commons and 1 peer reviewer at *eLife*, and the evaluation at *eLife* has been overseen by a Reviewing Editor and Anna Akhmanova as the Senior Editor.

Based on your manuscript, the reviews and your responses, we invite you to submit a revised version incorporating the revisions as outlined in your response to the reviews. Please note that all reviewers found several issues with consistency between data presented and written statements about the same data. Of course this can always happen in a manuscript, but please make sure in the revised manuscript that the data presented clearly supports your written statements and conclusions.

In addition, we invite you to consider these few additional comments from *eLife*.

1. A first concern is the absence of reference to important previous papers. Particularly, the work of Wiggins et al. in Dev. Cell 10.1016/j.devcel.2011.12.026 already reports aberrant bleb morphologies and actin accumulation under cofilin depletion in HeLa cells. Please cite this paper and modify the introduction to highlight the novelties of your study compared to previous works.

2. Another concern is about the Lifeact/photoconversion experiments, which should be interpreted with caution. Such experiments usually report more Lifeact's binding/unbinding dynamics than F-actin's turnover itself, and the very rapid (second-timescale) lifetime measured in these experiments is coherent with this possibility.

We appreciate that a significant difference is measured in the double siRNA condition, suggesting that both effects (actin turnover and Lifeact dynamics) may contribute in those experiments. Please discuss this point.

3. In the introduction, description of the biochemical effects of cofilin should go beyond citing one recent publication (Wioland et al.). For complete description, you should also mention the fact that cofilin's activity is highly dependent on its concentration, promoting severing at low concentration but stabilizing on the contrary actin filaments at high concentration (see papers from the Pollard, Blanchoin and De La Cruz labs). You should also mention the fact that cofilin is unlikely to function alone in cells, but together with catalyzing factors such as Aip1 or CAP (see for examples papers from the Michelot, Goode, Brieher and Lappalainen labs). Please discuss whether this complex behavior could change the interpretation of your results or not.

4. Following the previous comment, whether ADF and cofilin1 have overlapping functions or not (please consider citing also Hotulainen et al. Mol Biol Cell 2005) is complicated by the strong dependence of cofilin concentration on its activity. For some results (mainly in Figure 5), it is difficult to conclude whether ADF and cofilin1 have indeed different functions, or whether this is just a concentration effect.

Therefore, I would suggest 2 things. Would there be a way to compare the expression level of cofilin 1 and ADF1, so that one could evaluate properly the total level of expression of cofilin1 + ADF in each experiment. Then, could you try a rescue experiment, by expressing ADF in siCFL1 cells?

---

## [Author Response]

Based on your manuscript, the reviews and your responses, we invite you to submit a revised version incorporating the revisions as outlined in your response to the reviews. Please note that all reviewers found several issues with consistency between data presented and written statements about the same data. Of course this can always happen in a manuscript, but please make sure in the revised manuscript that the data presented clearly supports your written statements and conclusions.

We apologize for these errors, the manuscript text has now been reviewed for accuracy.

In addition, we invite you to consider these few additional comments from eLife.1. A first concern is the absence of reference to important previous papers. Particularly, the work of Wiggins et al. in Dev. Cell 10.1016/j.devcel.2011.12.026 already reports aberrant bleb morphologies and actin accumulation under cofilin depletion in HeLa cells. Please cite this paper and modify the introduction to highlight the novelties of your study compared to previous works.

We agree with the reviewer and previously cited the important work by Wiggan *et al.*, however, we had not discussed in significant detail the many differences between our works. Therefore, we have now added to the Introduction, paragraph 3:

‘Although Wiggan et al. has previously reported bleb defects in HeLa cells depleted of actin severing factors, the role of ADF and cofilin-1 in regulating the cortical actomyosin flow in fast amoeboid cells has not been determined (15).’

Additionally, to the Discussion, paragraph 2:

‘A similar elongated neck phenotype was observed by Wiggan et al., whereas they reported an increase in the frequency of HeLa cells with blebs after cofilin-1 RNAi (15). While these authors contend that cofilin-1 inhibits contractility by competing with myosin for F-actin binding, our results are in line with work showing that cofilin-1 increases contractility through optimizing actin filament lengths and de-branching (13,20,21).’

2. Another concern is about the Lifeact/photoconversion experiments, which should be interpreted with caution. Such experiments usually report more Lifeact's binding/unbinding dynamics than F-actin's turnover itself, and the very rapid (second-timescale) lifetime measured in these experiments is coherent with this possibility.We appreciate that a significant difference is measured in the double siRNA condition, suggesting that both effects (actin turnover and Lifeact dynamics) may contribute in those experiments. Please discuss this point.

This is a very good point, we have now qualified these data by adding to the Results section, ‘Rapid cortical actin flow requires ADF and cofilin-1 severing at leader bleb necks,’ paragraph 2:

‘However, as our measurements will be a composite of the actin turnover rate and the binding kinetics of LifeAct, these data should be strictly viewed as apparent rates of actin turnover.’

3. In the introduction, description of the biochemical effects of cofilin should go beyond citing one recent publication (Wioland et al.). For complete description, you should also mention the fact that cofilin's activity is highly dependent on its concentration, promoting severing at low concentration but stabilizing on the contrary actin filaments at high concentration (see papers from the Pollard, Blanchoin and De La Cruz labs). You should also mention the fact that cofilin is unlikely to function alone in cells, but together with catalyzing factors such as Aip1 or CAP (see for examples papers from the Michelot, Goode, Brieher and Lappalainen labs). Please discuss whether this complex behavior could change the interpretation of your results or not.

These are very good points, we have now discussed the dependence on cofilin-1 concentration and the possible contribution of functional interactions with Aip1 and CAP1 to our Introduction (paragraph 3) and Discussion (last paragraph) sections, respectively (citing work by Pollard, Blanchoin, and others).

4. Following the previous comment, whether ADF and cofilin1 have overlapping functions or not (please consider citing also Hotulainen et al. Mol Biol Cell 2005) is complicated by the strong dependence of cofilin concentration on its activity. For some results (mainly in Figure 5), it is difficult to conclude whether ADF and cofilin1 have indeed different functions, or whether this is just a concentration effect.

We have now discussed/cited Hotulainen, P *et al.* (2005) within the revised manuscript.

Therefore, I would suggest 2 things. Would there be a way to compare the expression level of cofilin 1 and ADF1, so that one could evaluate properly the total level of expression of cofilin1 + ADF in each experiment. Then, could you try a rescue experiment, by expressing ADF in siCFL1 cells?

These are very good points, we have found (1) by measuring endogenous mRNA levels, the relative level of cofilin-1 to be over 3-fold higher than ADF (see Figure 1—figure supplement 1B) and (2) in cells without both cofilin-1 and ADF, that reconstitution with cofilin-1 and not ADF will rescue cell compressibility (see figure 5B). Source data for figure 5A-B have also been provided with the revised manuscript. Together, these data further support the notion that cofilin-1 and ADF have non-overlapping roles during the fast amoeboid (leader bleb-based) migration of cancer cells.

From Review Commons:Reviewer #1 (Evidence, reproducibility and clarity (Required)):In this manuscript, Ullo and Logue investigate the roles of ADF and cofilin-1 in confined cell migration using A375 melanoma cells as a model. They found that depletion of both ADF and cofilin-1 suppressed the speed of bleb-based migration in confinement. Moreover, this dual intervention (1) resulted in cells displaying blebs with elongated necks, (2) reduced the bleb retraction rate, (3) increased the number of cortical barbed ends, (4) slowed down the rate of actin turnover at leader bleb necks and (5) impeded myosin minifilament flow rate after cofilin-1 inactivation. Overall, this is an interesting study worthy of publication pending appropriate revision.Comments:1. The authors have made some inaccurate statements throughout their manuscript. Specifically:a. In the abstract, they state that: RNAi of ADF and cofilin-1 led to a significant decrease in cell stiffness. Yet, Figure 5A shows that ADF depletion has no effect.

We apologize for the error, the abstract now correctly states that RNAi of ADF has no effect on cell stiffness.

b. On p. 6, "While the rate of actin turnover in cells depleted of cofilin-1 or ADF alone was modestly slower"…Yet, Figure 4G shows these individual knockdowns did not alter half-life.

We apologize for the error, the manuscript now correctly states that RNAi of cofilin-1 or ADF alone does not significantly alter the half-life of F-actin.

c. On p. 8, "depleting both proteins led to a large increase in the rate of actin turnover at bleb necks". Yet, Figure 4G shows the exact opposite (i.e., the half-life of F-actin turnover increased, which means that the turnover rate is reduced).

We apologize for the lack of clarity, the manuscript has been edited accordingly.

2. Does this Figure 1F represent data from one experiment? What is the S.D.?

Figure 1F represents pooled data (percentages) taken from 1E and G. Relevant statistics can be found in Figure 1G.

3. The authors should provide the blot showing the extent of cofilin-1 rescue.

Cell-by-cell analyses were used to quantify the relative levels of fluorescent protein (FP) tagged versions of cofilin-1 and ADF. More specifically, the level of cofilin-1 or ADF was determined for untransfected and transfected cells by immunofluorescence. Transfected cells were identified using the FP tag. Accordingly, the levels of FP tagged cofilin-1 and ADF were measured relative to the endogenous protein. Using this methodology, we were able to accurately determine that FP tagged versions of cofilin-1 and ADF were similar to endogenous. See figures 1I and 5C.

4. Figure 2A: what is the reason for showing cells at different time points? If the authors wish to show the retraction of a bleb(s), some cells at later time points go out of frame (e.g., non-target control).

Although I fully understand the reviewer’s point of view that these montages may not be necessary, we feel that these montages may help some readers to fully appreciate the range of observed phenotypes.

5. For the sake of completion, the authors should quantify the leader bled area and bleb area for siADF and dual KD in Figures 2C-D.These data have now been added. ADF RNAi was not found to significantly reduce either leader bleb or bleb (i.e., measuring all blebs) areas. Source data for leader bleb area and bleb area have also been provided with the revised manuscript.6. Figure 2I has no quantification. Could the authors provide quantification at the population level?

This quantification has now been provided. More specifically, by segmenting leader blebs into 10 different regions, we found cofilin-1 to be enriched near the leader bleb neck in 10 cells.

7. In Figure 3D (as well as other figures), it is not clear what groups are being compared statistically.

In Figure 3D, groups are being compared to cells before reconstitution. In similar graphs, groups without an “ns” or stars are the comparison group.

8. On p. 6, the authors state that "in cells depleted of ADF and cofilin-1, transfection of EGFP-cofilin-1 was not sufficient to restore cortical barbed ends to a level similar to control (Figure 3H). Either their statement is incorrect or "ns" in Figure 3H compares cofilin WT to EGFP alone (which is misleading).

We thank the reviewer for pointing out this error, our statement was incorrect and has been corrected to:

‘However, in cells depleted of ADF and cofilin-1, transfection of EGFP-cofilin-1 was sufficient to restore cortical barbed ends to a level similar to control (Figure 3H).’

9. What is the rationale for performing experiments with trypsinized cells? Are the phenotypes and underlying molecular pathways same for compressed and trypsinized cells ?

Because of the nature of our method for cell confinement, many assays that employ chemical treatments or the preparation of cell lysates are unavailable. Therefore, we use freshly trypsinized (spherical) cells, which predominantly have cortical actin and that are amenable to a unique range of assays, to in general study the cortical actin network. Previous work by us (Logue *et al.* 2015, *eLife*) and others (Bergert *et al.* 2012, *PNAS* and Liu *et al.* 2015, *Cell*) have shown that the properties of the cortical actin network in spherical cells are predictive of whether or not a cell will undergo fast amoeboid (leader bleb-based) migration.

10. Is the reduction in myosin flow (Figure 5F) due to reduced actin turnover? What is the role of myosin in the neck formation ? From the proposed model in Figure 6, it looks like the neck is formed by myosin based contractility. Will inhibition of myosin activity by blebbistatin reduce the length of the neck ?

We will address each question in turn; (1) correct, our model is that actin turnover at the bleb neck is essential for sustained myosin contractility and thus, myosin flow, (2) we and others believe that myosin is essential to the formation of the constricted neck, however, the molecular basis for why myosin is enriched at bleb necks is not well understood, and (3) this is a very interesting idea, we would also predict that the length of the neck would be reduced upon blebbistatin treatment. Unfortunately, this would be difficult to test in cells, as high myosin activity is required to initiate a bleb.

11. The authors should do justice and cite relevant articles which show that confinement induces cell blebbing: PMID: 32789173 and PMID: 31690619

These articles are now discussed/cited within the revised manuscript.

Reviewer #1 (Significance (Required)):Following appropriate revision and clarifications, the resulting manuscript will represent a significant contribution to the area of cell motility.Referees cross-commentingReviewer #2 and I are in agreement. I also concur with her/his point (which I did not include in my review) about the use of two different siRNAs and inclusion of data on cell directionality. Lastly, I was "generous" regarding the estimated time to complete revisions because the π is a young Assistant Professor, and I just wanted to eliminate the pressure associated with time constraints so that he revises the manuscript meticulously.

We wish to thank the reviewer for their comments, which have undoubtedly improved our manuscript.

Reviewer #2 (Evidence, reproducibility and clarity (Required)):This paper interrogates the function of ADF-1 and cofilin in the leader-bleb based migration, as observed primarily in melanoma cells in confinement. Overall, this paper is reasonable and the data generally supports the conclusions, though robust statistical analysis is lacking in places. The experiments appear to be carefully performed and the FLAP experiments are innovative and informative. The main findings are that ADF-1 and cofilin each have roles in the phenotype of cells that undergo bleb-based migration and the results of inhibiting either one are distinct from (and milder than) those of inhibiting both.Specific Comments1. When the authors switch from studying cells with obvious leader based blebs to ones that are "spherical" it is unclear what else changes: is the confinement different? Are these migratory cells, etc?

Because of the nature of our method for cell confinement, many assays that employ chemical treatments or the preparation of cell lysates are unavailable. Therefore, we use freshly trypsinized (spherical) cells, which predominantly have cortical actin and that are amenable to a unique range of assays, to in general study the cortical actin network. Previous work by us (Logue *et al.* 2015, *eLife*) and others (Bergert *et al.* 2012, *PNAS* and Liu *et al.* 2015, *Cell*) have shown that the properties of the cortical actin network in spherical cells are predictive of whether or not a cell will undergo fast amoeboid (leader bleb-based) migration.

2. In at least some cases, the figures are not labeled and described (in captions and text) well enough to help the reader understand their import. For example, Figure 1B is hardly described in either the caption or the text. In Figure 1e there are no axis labels. Are those microns? While the authors say how the cells were selected as "highly motile" in the Methods section it would be convenient if this was restated in the figure caption.

We apologize for the lack of clarity, the manuscript has been edited accordingly.

3. An additional concern about Figure 1 is as follows: the authors state that ADF KD results in a decrease in mobility but that statement is not accurate given the data they show, where ADF is not statistically different than control and actually has the cell with the highest average speed (although it is possible that the highest cell both in ADF and in the control are statistical outliers and should be removed from the datasets. The authors should conduct a Grubs test to confirm whether they should keep these data points in their analysis). Additionally, generally when people are discussing changes in mobility they are looking not just at the speed but also at the directionality and directional persistence of cells. It would be interesting to present this data here as well and it simply requires analysis of tracks the authors already have in hand.

We apologize for the lack of clarity, the manuscript has now been edited to more accurately state:

‘Depleting cells of ADF led to a slight reduction in the number of highly motile cells (Figure 1E-H).’

As a general practice, we don’t remove any cells from our migration analyses. This is because it is not uncommon to observe a wide range of migration parameters under conditions of confinement.

Directional autocorrelations have now been provided. Please note that these autocorrelations were only done for highly motile cells, as directionality measurements on non- or poorly moving cells (e.g., moving in place) would not be informative. Accordingly, conditions with few motile cells will have a relatively low n-value. See Figure 1—figure supplement 1C.

4. The data presented in Figure 2A is intriguing but also points to difficulty of concluding that the effects of ADF and cofilin are "additive." At times the authors use "non-overlapping," which I believe is more accurate given the results presented in Figure 2 and elsewhere in the manuscript. The blebs apparent in the dual knockdown are wild-looking and distinct from those seen in any of the other panels. The authors hardly discuss this even though they feature the elongated neck phenotype and highlight that it is not present in other treatments. If the effect of ADF and cofilin are "additive" one would think you would have a similar phenotype but featuring, for example, short necks in each case and then a long one when both are disrupted. At the very end of the paper the authors discuss that cofilin-1 is required for actin turnover and myosin contractility at the bleb neck. They have a very brief statement that KD of both proteins leads to an elongated bleb neck, but then spend the rest of the paragraph focusing on the effects of just cofilin-1. I think that investigating the long-necked blebs further would be of interest, especially since these cells continue to be motile and without any clear "leader bleb". Is actin present in these blebs? What is the flow like? How does this direct motility (and directionality) in these cells. Investigation of this morphology may lead to a better understanding of exactly how cofilin-1 and ADF interact to regulate blebbing and migration, as only lack of both proteins results in this altered morphology.

We apologize for the confusion, 25/79 cells display the severe elongated neck phenotype (Figure 2D). In cells lacking both cofilin-1 and ADF, actin (Figure 3A) and myosin (Figure 5D) accumulate at bleb necks. In the absence of sufficient turnover, cortical actin flow is impeded (Figure 4D) and thus, these cells do not undergo leader bleb-based migration. Accordingly, these data help to form the basis of our model in which ADF and cofilin-1 collaborate to promote actomyosin flow and leader bleb-based migration (Figure 6).

5. It is generally expected that authors will try two different siRNAs (or one siRNA and a function-inhibiting antibody) to provide reassurance that effects are on-target.

In general, we agree with the reviewer’s point of view, however, in this paper we have used RNAi, reconstitution, and CALI in order to provide reassurance that effects are on-target.

6. The authors state, "ADF and cofilin-1 rapidly disassemble cortical actin" and the authors discuss a change in the level of F-actin. However, it is not clear whether there is an overall change in actin level in the KD cells or if there is (as suggested) simply a change in ratio of F-actin : G-actin, which implies two rather different functions of these proteins.

We apologize for the confusion, as we use phalloidin, our assays specifically measure changes in the level of F-actin.

7. It would be very interesting indeed if the authors could test their speculation about uncontrolled filament elongation is contributing to the unusual phenotypes observed.

In future work, which is likely to require purified proteins in actin polymerization and other assays (beyond the scope of the current work), we will aim to test the idea that uncontrolled actin polymerization contributes to the observed phenotypes.

Reviewer #2 (Significance (Required)):General interest level in leader bleb-based migration is not especially high due to the fact that it is provoked by (non-physiological) confined environments. Even without direct physiological relevance, migratory strategies that cancer cells may adopt may be of interest to a broad community and the findings here (of "additive" effects of ADF and cofilin-1) may have relevance beyond this particular migratory mode. The main weaknesses of the manuscript is the incomplete description of the additive affect of ADF and cofilin: while the authors attempt to capture this in their discussion and schematic depiction, how the two proteins interact with respect to leader based bleb migration and related processes such as actin turnover remain incompletely described.

We would note that our assay was designed to simulate tissue environments in which cells are under conditions of high mechanical confinement. Work by us and others has shown that these assays efficiently promote the transition to fast amoeboid (leader bleb-based) migration, similar to what has been previously observed in vivo (Tozluoglu *et al.* 2013, *Nature Cell Biology*, Ruprecht *et al.* 2015, *Cell* and Venturini *et al.* 2020, *Science*).

We in general agree with the reviewer’s point of view that much remains to be done with regards to understanding the non-overlapping roles of cofilin-1 and ADF during leader bleb-based migration. However, we view this paper as providing the rationale for future (in-depth) biochemical studies. As mentioned in our point-by-point responses (above), this work is likely to involve using purified proteins in actin polymerization and other assays, which we feel is beyond the scope of the current paper.

The main audience for this work will be anyone interested in novel migratory strategies in cancer cells as well as those who wish to understand origins of cell motility from a biophysical point of view.Reviewer expertise is in novel migratory strategies of cancer cells, particularly in three-dimensional in vitro environments.Referees cross-commentingI think the other reviewer and I are generally in agreement. There are some interesting things in this manuscript but there are inconsistencies between written statements and data presented in places as well as lack of robust statistical analysis. Certainly, all data presented in figures and written description of such should be reconciled.

We apologize for these errors, as discussed in our point-by-point responses (above), these issues have now been reconciled.

The other question brought up by both reviewers that must be addressed is the fact that the authors switch from studying confined to trypsinized spherical cells at some point. Why should these two situations be studied side-by-side? There is an implication that the same underlying molecular pathways are relevant, but what evidence is there that this is the case?

We have copied our response to specific comment 1 (reviewer 2):

Because of the nature of our method for cell confinement, many assays that employ chemical treatments or the preparation of cell lysates are unavailable. Therefore, we use freshly trypsinized (spherical) cells, which predominantly have cortical actin and that are amenable to a unique range of assays, to in general study the cortical actin network. Previous work by us (Logue *et al.* 2015, *eLife*) and others (Bergert *et al.* 2012, *PNAS* and Liu *et al.* 2015, *Cell*) have shown that the properties of the cortical actin network in spherical cells are predictive of whether or not a cell will undergo fast amoeboid (leader bleb-based) migration.